# Whole blood immunophenotyping uncovers immature neutrophil-to-VD2 T-cell ratio as an early marker for severe COVID-19

Guillaume Carissimo [1,2,15✉], Weili Xu[2,15], Immanuel Kwok[2,15], Mohammad Yazid Abdad [3], Yi-Hao Chan [1,2], Siew-Wai Fong[1,2,4], Kia Joo Puan [2], Cheryl Yi-Pin Lee[1,2], Nicholas Kim-Wah Yeo[1,2], Siti Naqiah Amrun[1,2], Rhonda Sin-Ling Chee[1,2], Wilson How[2], Stephrene Chan[5,6,7,8], Bingwen Eugene Fan [5,6,7,8], Anand Kumar Andiappan [2], Bernett Lee[2], Olaf Rötzschke[2], Barnaby Edward Young [3,9,10], Yee-Sin Leo [3,9,10,11,12], David Chien Lye[3,9,10,11], Laurent Renia [1,2], Lai Guan Ng [2], Anis Larbi[2] & Lisa FP Ng [1,2,13,14✉]

SARS-CoV-2 is the novel coronavirus responsible for the current COVID-19 pandemic. Severe complications are observed only in a small proportion of infected patients but the cellular mechanisms underlying this progression are still unknown. Comprehensive flow cytometry of whole blood samples from 54 COVID-19 patients reveals a dramatic increase in the number of immature neutrophils. This increase strongly correlates with disease severity and is associated with elevated IL-6 and IP-10 levels, two key players in the cytokine storm. The most pronounced decrease in cell counts is observed for CD8 T-cells and VD2 γδ T-cells, which both exhibit increased differentiation and activation. ROC analysis reveals that the count ratio of immature neutrophils to VD2 (or CD8) T-cells predicts pneumonia onset (0.9071) as well as hypoxia onset (0.8908) with high sensitivity and specificity. It would thus be a useful prognostic marker for preventive patient management and improved healthcare resource management.

[1] Infectious Disease Horizontal Technology Center, Agency for Science, Technology and Research, Immunos, Biopolis 138648, Singapore. [2] Singapore Immunology Network, Agency for Science, Technology and Research, Immunos, Biopolis 138648, Singapore. [3] National Centre for Infectious Diseases, 16 Jalan Tan Tock Seng, 308442 Singapore, Singapore. [4] Department of Biological Sciences, National University of Singapore, Singapore, Singapore 117543. [5] Department of Haematology, Tan Tock Seng Hospital, 11 Jalan Tan Tock Seng, 308433 Singapore, Singapore. [6] Department of Laboratory Medicine, Khoo Teck Puat Hospital, Singapore, Singapore. [7] Lee Kong Chian School of Medicine, Singapore, Singapore. [8] Yong Loo Lin School of Medicine, Singapore, Singapore. [9] Department of Infectious Diseases, Tan Tock Seng Hospital, 11 Jalan Tan Tock Seng, 308433 Singapore, Singapore. [10] Lee Kong Chian School of Medicine, Nanyang Technological University, 11 Mandalay Road, 308232 Singapore, Singapore. [11] Yong Loo Lin School of Medicine, National University of Singapore and National University Health System, 10 Medical Drive, 117597 Singapore, Singapore. [12] Saw Swee Hock School of Public Health, National University Singapore, 12 Science Drive 2, 117549 Singapore, Singapore. [13] Department of Biochemistry, Yong Loo Lin School of Medicine, National University of Singapore, 8 Medical Drive, 117596 Singapore, Singapore. [14] Institute of Infection, Veterinary and Ecological Sciences, University of Liverpool, Liverpool, 8 West Derby Street, Liverpool L7 3EA, UK. [15] These authors contributed equally: Guillaume Carissimo, Weili Xu, Immanuel Kwok. ✉email: guillaume_carissimo@immunol.a-star.edu.sg; lisa_ng@immunol.a-star.edu.sg

Severe Acute Respiratory Syndrome coronavirus 2 (SARS-CoV-2) first appeared in Wuhan, China in late 2019. It is a novel pathogen responsible for the coronavirus disease 2019 (COVID-19) pandemic[1]. COVID-19 patients experience a wide spectrum of clinical manifestations that ranges from low-grade fever and mild respiratory symptoms, to more severe forms. This includes acute respiratory distress syndrome (ARDS), which requires provision of supplemental oxygen, and in some cases intubation and mechanical ventilation[2–5]. The majority of critical cases of COVID-19 are associated with coagulopathy with a high prevalence of thromboembolic events in patients under mechanical ventilation which lead to inclusion of anticoagulation therapies in the standard of care of severe COVID-19 cases[6–8]. While the strong inflammatory response to COVID-19 is proposed to be associated to COVID-19 associated coagulopathy[6], it remains unclear how SARS-CoV-2 infection affects the activation of immune cells and their contribution towards the severity of disease outcomes in patients.

Previous clinical studies reported associations with clinical blood counts, while others specifically assessed T-cell subsets for activation and exhaustion markers[9–11]. Since strong evidence points to a cytokine storm as the culprit for disease severity[12,13], various groups have investigated cytokine-secreting pathogenic T-cells and inflammatory monocytes that could have triggered this phenomenon[9–11]. In addition, flow cytometry analysis in COVID-19 patients has also shown a polarisation towards the Th17 subtype and a highly activated and exhausted CD8[+] T-cell compartment[14,15]. All these studies were carried out on peripheral blood mononuclear cells (PBMCs), thus excluding most granulocyte populations[14,15]. However, to elucidate all the immune subsets that could potentially trigger severe COVID-19 pathology, it is imperative to perform comprehensive whole blood immunophenotyping of COVID-19 patients which includes granulocyte populations.

In this study, we employ high dimensional flow cytometry to analyse a wide spectrum of more than 50 subsets of the myeloid and lymphoid immune cell compartments. The study focuses on a cohort of 54 COVID-19 patients who presented with varied clinical manifestations ranging from mild to fatal outcomes during the ongoing SARS-CoV-2 pandemic in Singapore. This comprehensive immunophenotyping identifies immature neutrophils, CD8 T-cells and gamma delta (VD) 2 T-cells as key immune cell populations that undergo substantial changes in the cell counts across the spectrum of clinical severity. Their numbers, in fact, represent an early and robust prognosis value as shown by receiver operating characteristics (ROC) analysis.

## Results

**Circulating myeloid cells are decreased in COVID-19 patients.** A total of 54 patients with laboratory-confirmed SARS-CoV-2 infection were recruited at the National Centre for Infectious Diseases (NCID), Singapore from end March to mid-May 2020 (Supplementary Table 1). Blood was collected from 54 patients upon enrolment at a median 7 days post-illness onset (pio) (Supplementary Table 1), from 28 patients who had recovered from COVID-19 disease (median 30 days pio) (Supplementary Table 1) and 19 healthy donors (Supplementary Table 2). Unfortunately, only 11 patients had paired acquisition between acute and recovered which prevented meaningful paired analysis (Supplementary Table 1). Immunophenotyping of whole blood samples was carried out with three distinct flow cytometry panels to analyse myeloid, granulocyte and lymphoid subsets. (Fig. 1a, Supplementary Table 3). Each panel was supplemented with counting beads to allow accurate assessment of cell counts. 19 of the 54 acute patients had paired plasma samples that permitted

quantification of immune mediators by Luminex multiplex microbead-based immunoassay. The cohort was strongly biased towards males, of which three patients experienced a thromboembolism event (5.6%) and two patients had fatal outcomes (3.7%).

First, we assessed using healthy donor samples, if the different blood collection method for recovered samples affected cell counts or activation markers. We observed that, while the cell count was not affected, expression of activation markers was affected on most cells but not CD38+ on T-cells (Supplemental Fig. 1a), allowing direct comparison of activation markers only between healthy and acute samples. The FACS analysis revealed a declined cell count for eosinophils, basophils, total T-cells, dendritic cells (DCs), natural killer (NK) CD56 Bright, and plasmacitoid DCs (pDCs) in patients with acute COVID-19 infection (Fig. 1b, Supplementary Fig. 1a). No significant changes were observed for B-cells, total monocytes, and total NK cells (Fig. 1b, Supplementary Fig. 1a). Unbiased analysis by Uniform Manifold Approximation and Projection (UMAP) and graph-based clustering however identified with CD169+ monocytes and CD11b[high] neutrophils, two additional clusters with high variation in acute patients (Fig. 1c). Further analysis showed that the monocytes presented with an increased expression of CD169 (strong type I interferon signature marker[16]), increased expression of CD11b and HLA-DR, as well as CD33, a constitutive PI3K signalling inhibitor[17,18] (Fig. 1d, Supplementary Fig. 1b) as compared to healthy donors.

Similar to the monocytes, neutrophils showed a significant upregulation of CD11b, CD66b, Siglec 8, CD38 and HLA-DR, suggesting that they were activated in response to SARS-CoV-2 infection (Fig. 1e, Supplementary Fig. 1c). Interestingly, despite this activation phenotype, an increase in the overall number of circulating neutrophils during acute SARS-CoV-2 infection based on conventional phenotypic markers (CD66b, CD11b and CD16) was observed only in a small subset of our cohort (Fig. 1f). However, in-depth analysis of neutrophil subsets allows discrimination between immature (CD16[low/high]CD10[−]) and mature (CD16[high]CD10[+]) subsets (Fig. 1g)[19–21]. Overall, a significant increase of immature neutrophil numbers was observed in acute patients as compared to healthy donors or recovered patients, while the number of mature neutrophils decreased (Fig. 1h).

**CD8 and γδ T-cells are the most affected lymphocyte subsets.** To better characterise COVID-19-induced lymphopenia, levels of CD8, CD4, γδ (i.e., VD1 and VD2), and mucosal-associated invariant T-cells (MAIT, CD3[+]VA7.2[+]CD161[+]) were assessed during acute infection. Results showed a decrease in circulating MAIT, CD8[+] and VD2 T-cells (Fig. 2a). However, circulating VD1 T-cells did not vary in numbers, and CD4[+] T-cells did not show a significant decrease during acute infection (Fig. 2a). Interestingly, levels of regulatory T-cells (Treg) and CD4[+]CD161[+] T-cells increased in recovered patients as compared to acute patients (Fig. 2a).

Next, UMAP analysis was done on CD3[+] cells to visualise changes in differentiation states within the T-cell compartments (Fig. 2b). UMAP visualisation suggests that phenotypic modulation in the CD8[+] cluster was the most pronounced during SARS-CoV-2 infection (Fig. 2b). In order to validate this observation, CD45RA and CD27 markers were used to analyse the frequency of naïve (CD45RA[+]CD27[+]), central memory (CM, CD45RA[−]CD27[+]), effector memory (EM, CD45RA[−]CD27[−]) and terminal effector (TEMRA, CD45RA[+]CD27[−]) amongst the T-cell populations (Fig. 2c, Supplementary Fig. 2). In agreement with the UMAP analysis, CD8[+] T-cells showed a change in differentiation

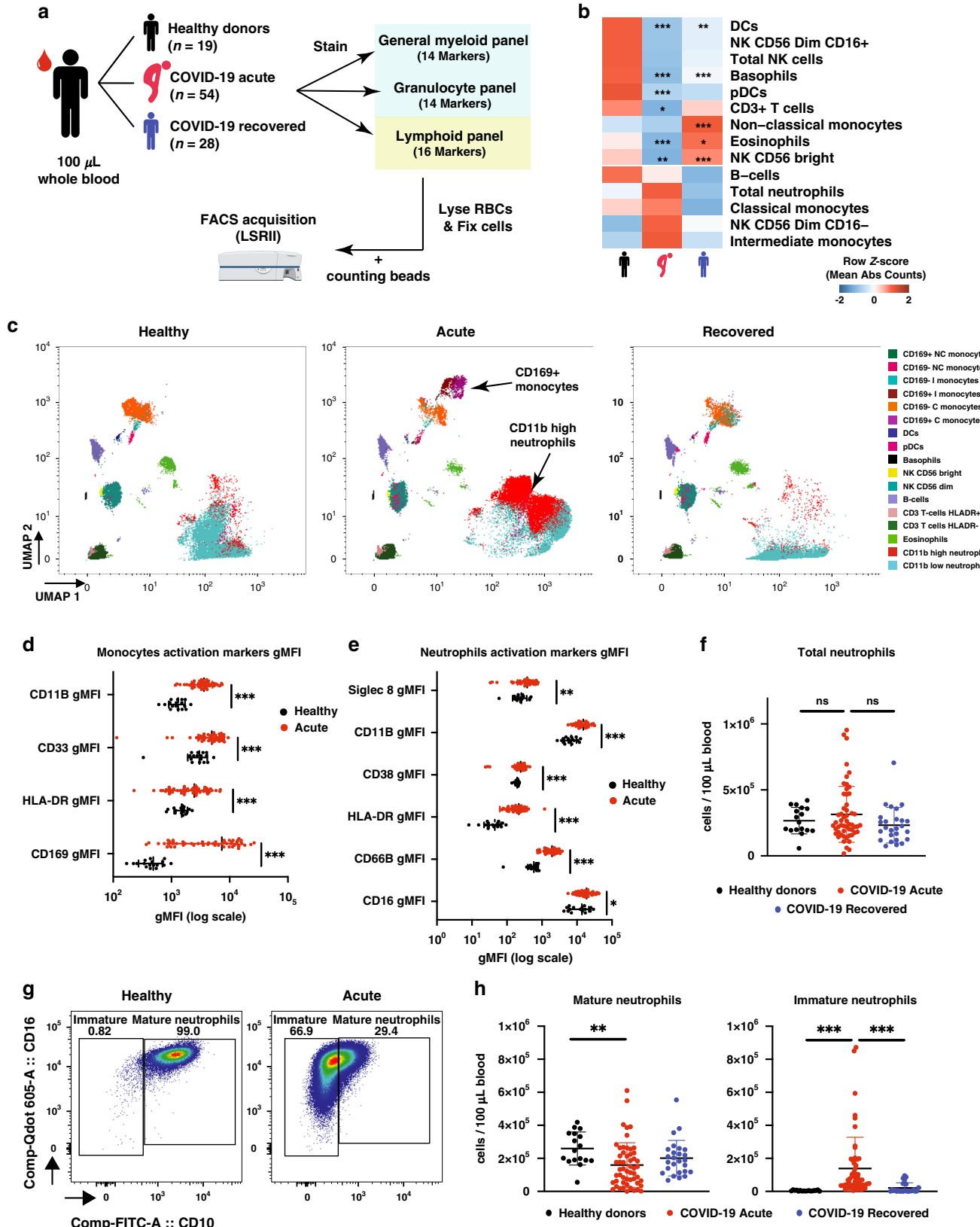

profile from naïve in favour of EM and TEMRA (Fig. 2c). Noticeably, the frequency of naïve CD4$^+$ T-cells decreased but was not reflected in a significant increase of a specific differentiated population (Fig. 2c).

In addition, UMAP analysis also suggested changes in VD1 and VD2 populations that were not reflected in terms of

differentiation (Fig. 2b, c). Therefore, we investigated the expression of general activation marker CD38. In this context, we observed that all differentiation stages of CD8$^+$ T-cells, VD1 and VD2, had higher expression of CD38 except VD2 TEMRA (Fig. 2d). On the other hand, CD4$^+$ T-cells did not show difference in the CD38 activation marker expression

**Fig. 1 SARS-CoV-2 infection induces a decrease in immune cells in peripheral blood. a** Schematic representation of flow cytometry workflow. **b** Heatmap representation of row z-score of mean absolute cell counts across the groups. Individual plots are shown in Supplementary Fig. 1A. **c** UMAP clustering of CD45+ immune cells. **d** Monocyte activation markers mean geometric MFI (gMFI). **e** Neutrophil activation markers mean geometric MFI (gMFI). **f** Absolute neutrophil counts. **g** Representative plot of mature and immature neutrophil gating strategy in healthy control or acute COVID-19 patient. **h** Mature (CD10+) and immature (CD10−) neutrophil Abs counts. Data presented are from individual human samples of healthy $n = 17$, acute $n = 54$ and recovered $n = 26$ common in flow panels **a** and **c**. Heatmap is presented as mean of z-score, scatter dot plots are presented with mean ± SD. Absolute counts were analysed by Kruskal–Wallis using Dunn correction for multiple comparison, gMFI was analysed by Brown-Forsythe and Welch ANOVA without multiple comparison. For heatmap, stars shown in acute column represent healthy vs acute comparison. Stars shown in recovered column represent acute vs recovered comparison. ns non-significant. *$p < 0.05$, **$p < 0.01$, ***$p < 0.001$. Data available in source data file, exact $p$-values are given in Supplementary Data 1.

(Fig. 2d). Together, our data suggest that while circulating cell counts were generally decreased for T-cells, SARS-CoV-2 differentially impacts the different T-cell subsets in terms cell counts, differentiation and expression of CD38.

**Clinical severity is reflected by immune cell counts**. In order to associate the data with the clinical severity we separated the patients into four different groups: no pneumonia, pneumonia only, pneumonia and hypoxia, and pneumonia and hypoxia requiring ICU admission (Fig. 3a)[22,23]. This allowed estimation of cell counts in those groups and identification of markers that potentially depict disease severity. Consistent with previous studies on CD4 and CD8 lymphopenia[9,24,25], CD8+, CD4+, MAIT, VD1 and VD2 T-cells showed a gradual reduction in the peripheral blood with increasing disease severity (Fig. 3b). The effect was more pronounced for CD8+ and VD2 T-cells (Fig. 3b), suggesting a strong activation and infiltration of these cells in the lungs.

Cell counts in various myeloid subsets showed a similar decreasing profile with severity for pDCs, DCs, classical and intermediate monocytes (Fig. 3c). In contrast to cell counts, myeloid activation markers showed differential trends with severity (Fig. 3d). CD86 expression on DCs, HLA-DR and CD33 expression on monocytes followed a gradual decrease with increasing severity (Fig. 3d). Expression of CD169 on monocytes was decreased in ICU patients, while CD86 expression on pDCs was consistent across severity groups (Fig. 3d). Together, these results suggest that the remaining circulating monocytes and DCs in severe cases have a dysregulated phenotype.

While total circulating neutrophils showed no significant change with disease severity, neutrophilia was only observed in some patients with severe clinical complications (Fig. 3e). Particularly, there was a change in the composition of neutrophil subsets in accordance to disease severity, where an increase in the immature neutrophil cell count and frequency was accompanied with a decrease of mature neutrophils (Fig. 3e). These results suggest that immature neutrophils could reflect disease severity much more accurately than total neutrophil counts.

**Immature neutrophil absolute count correlates with cytokines**. Neutrophil-to-Lymphocyte Ratio (NLR) or Neutrophil-to-CD8 T-cell Ratio (N8R) were proposed to be good diagnostic and prognostic markers for severe COVID-19 respiratory disease[25,26]. However, these studies observed increased neutrophils in severe cases which was not consistent with our observations and in another study[27] (Figs. 1g and 3e). To validate that the identified populations would be good markers of disease severity, a correlation analysis between analyte levels in available paired plasma samples (from CPT sodium citrate tubes) was performed with the cell counts obtained in FACS (from EDTA vacuette tubes) (Fig. 4a, Supplementary Fig. 3). Interestingly, strong correlation scores were observed between analytes and immature neutrophil counts (Fig. 4a, Supplementary Fig. 3a), rather than with total neutrophil counts (Fig. 4a, Supplementary Fig. 3b). The strongest correlations were observed between immature neutrophil counts and IL-6 (rho = 0.6747, $p = 0.0015$), and IP-10 (rho = 0.7596, $p = 0.0002$) (Fig. 4b).

In addition, strong correlations were also observed between mature neutrophils, monocytes and intermediate monocytes, as well as CD8 and VD2 T-cell counts (Supplementary Fig. 3c). These results suggest that immature neutrophils counts can potentially be used as sensitive and reliable indicators of disease severity.

**Immature neutrophil/VD2 ratio an improved prognostic marker**. We next assessed if an immature neutrophil-to-CD8 T-cells ratio (iN8R) or VD2 T-cell counts ratio (iNVD2R) could be a better prognostic marker of disease severity as compared to the current proposed NLR and N8R[25,26]. To differentiate patients with and without pneumonia, iNVD2R performed better than N8R or iN8R with an area under receiver operating characteristic (AUROC) curve of 0.8451 (95% confidence interval CI: 0.7379–0.9523) vs 0.806 (95% CI: 0.6911–0.9210) and 0.7158 (95% CI: 0.5754–0.8562) respectively (Fig. 5a). In addition, to differentiate patients with and without hypoxia, an AUROC of 0.9111 (95% CI: 0.8306–0.9916) was obtained for iNVD2R as compared to 0.8931 (95% CI: 0.8044–0.9817) for iN8R and 0.7958 (95% CI: 0.6781–0.9136) for N8R. These results indicate that iNVD2R and iN8R could be good markers for severe respiratory disease.

To assess if this analysis could have predictive prognostic value in hospitalisation settings to improve patient management, we repeated the same analysis with only samples that were acquired at or before 7 days pio amongst the 54 acute patients (24 patients, median pio = 3 days, range 1 to 7 days pio). AUROC for iNVD2R showed strong prognostic value for pneumonia onset (0.9071) as well as for onset of hypoxia (0.8908) (Fig. 5b, Table 1). Our data show that immature neutrophil counts are better in predicting disease severity as compared to total neutrophil counts. Importantly, they can be used in a ratio with CD8 or VD2 lymphocyte counts to improve the current N8R predictive ratio.

**Discussion**

In this study, immunophenotyping of peripheral blood from COVID-19 patients revealed a significant shift in the ratio between mature and immature neutrophils associating with severity. The increased numbers of immature neutrophils and the disappearance of mature neutrophils likely reflect gradual and sustained mobilisation of these cells into the lungs in response to an ongoing inflammation, leading to premature release of immature neutrophils from the bone marrow[21]. Supporting this hypothesis, a recent study, investigating several myeloid populations between circulating PBMCs and the lung lavage of COVID-19 patients showed that granulocytes represent up to 80% of total

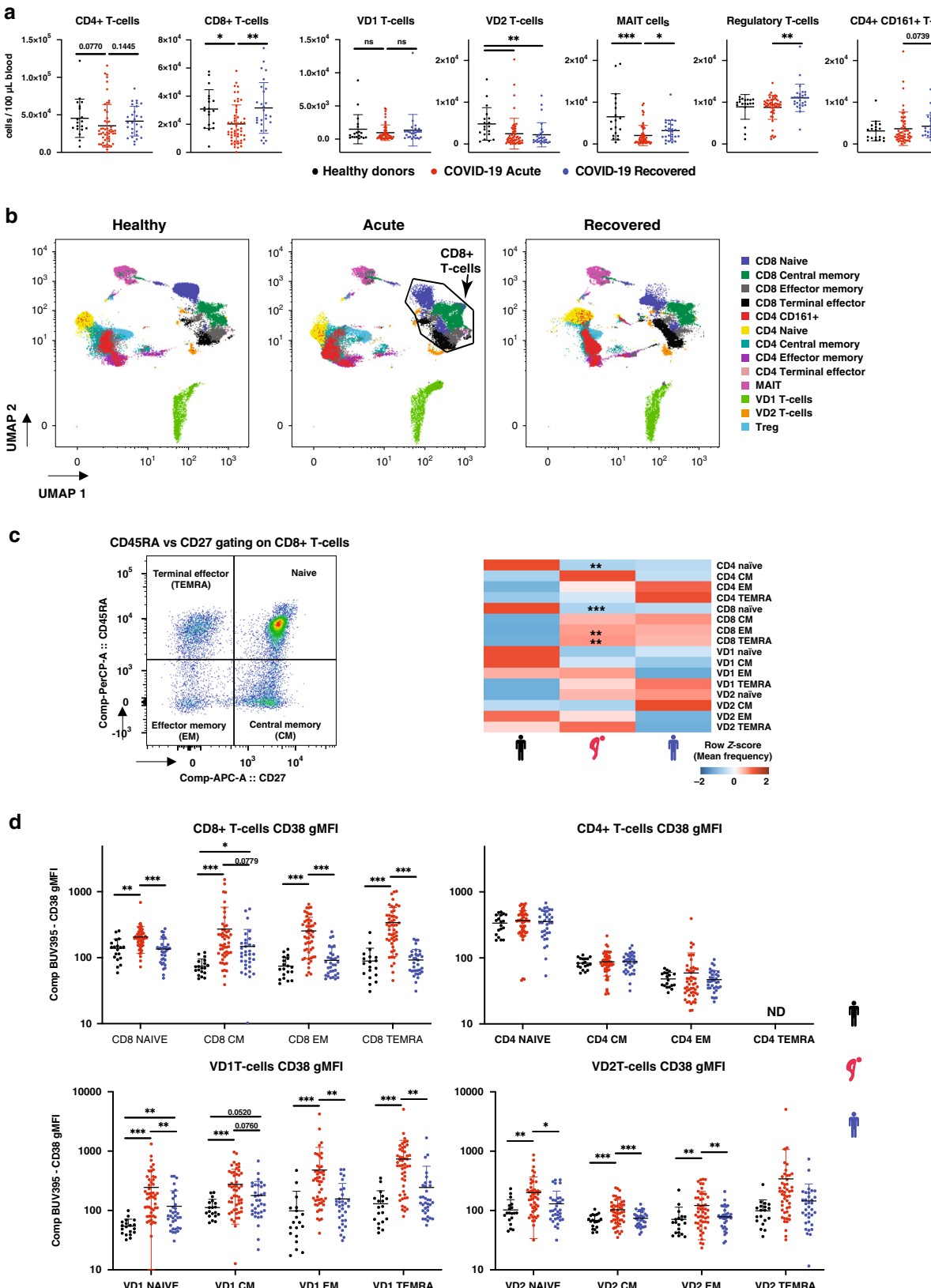

CD45[+] lung infiltrates[28]. In addition, autopsies of COVID-19 fatalities showed typical lesions associated with toxic neutrophil effects[29,30]. In line with this observation, marked morphological abnormalities of the circulating neutrophils were reported in COVID-19 patients[27]. These cells present typical hallmarks of immature neutrophils and their precursors such as band shaped nuclei and a lower expression of CD10 and CD16[31]. Consistent with our data, a recent non peer reviewed study on a small number of patients reported that the presence of low density inflammatory neutrophils was strongly associated with disease severity and IL-6 levels[32]. Functionally these low density neutrophils showed spontaneous extracellular trap formation,

**Fig. 2 SARS-CoV-2 infection induces general lymphopenia and CD8, VD1 and VD2 activation. a** Absolute counts of T-cell compartments in healthy donors, acute and recovered COVID-19 patient. **b** UMAP clustering of CD3+ cells. **c** left panel: CD45RA and CD27 gating strategy example on CD8+ T-cells; right panel: heatmap representation of mean frequencies of T-cell differentiation across the groups, individual plots given in Supplementary Fig. 2. **d** Changes in CD38 gMFI in naïve, CM, EM and TEMRA for CD8, CD4, VD1 and VD2 T-cells. ND indicates not determined since frequency of these compartment was too low for accurate gMFI measurement. Data presented are from individual human samples of healthy $n = 19$, acute $n = 54$ and recovered $n = 28$ from flow panel B. Absolute counts and frequency were analysed by Kruskal–Wallis using Dunn correction for multiple comparison, gMFI was analysed by Brown–Forsythe and Welch ANOVA using Dunnett T3 correction for multiple comparison. Scatter dot plots are presented as mean ± SD. For heatmaps, stars shown in acute column represent healthy vs acute comparison. Stars shown in recovered column represent acute vs recovered comparison. *$p < 0.05$, **$p < 0.01$, ***$p < 0.001$. Data available in source data file, exact $p$-values are given in Supplementary Data 1.

enhanced cytokine production and associated with D-dimer and systemic IL-6 and TNF-α levels[32]. We hypothesise that the CD11b[int]CD44[low]CD16[int] low density neutrophil population indentified in that study is likely constituted primarily of CD10− immature neutrophils. More recently, two studies used flow cytometry, single cell sequencing and mass cytometry to confirm the immature and dysfunctional phenotype in the myeloid populations, including these neutrophils[33,34]. Interestingly, the diagnostic value of a neutrophil left shift (banded versus segmented neutrophils) had previously been explored in order to predict infectious diseases in addition to inflammatory diseases[35] and is therefore not limited to COVID-19 severity. Similarly, the presence of immature low density neutrophils have been reported in the literature for various infectious and inflammatory diseases[36–38] as well as induced by LPS in healthy subjects[39], highlighting the necessity of future studies to compare the role and function of these COVID-19 immature neutrophils with the circulating immature neutrophils present in other diseases.

During SARS-CoV-2 infection, immature neutrophil numbers strongly correlated with IL-6 and IP-10. IL-6 and IP-10 are consistently upregulated during a cytokine storm and are associated with severe ARDS[9,12,40,41]. While some studies report inflammatory monocytes as the source of IL-6[9,42,43], our results suggest that immature neutrophils could also be a non-negligible source of IL-6 during COVID-19-induced cytokine storm. Indeed, neutrophils have been found to produce biologically relevant amounts of IL-6 after engagement of TLR8, a toll like receptor recognising single strand RNAs of viral or bacterial origin[44,45]. Since IL-17 operates upstream of IL-1 and IL-6, and is a major orchestrator of sustained neutrophils mobilisation[46], it is plausible that IL-17 could significantly affect the neutrophils compartment in COVID-19 patients. Consistent with this hypothesis, CD4 T-cells in COVID-19 patients are skewed towards a Th17 phenotype[15], and we also observed increased CD4+CD161+ T-cells in recovered patients. These CD4+CD161+ T-cells are known to be either IL-17 producer cells or their precursors[47]. Thus, our results could reflect the re-circulation of these cells from the lung or secondary lymphoid organs after infection and support the possibility of IL-17 in mediating neutrophil damage to the lungs. Together, this would support proposed anti-IL-17 or JAK2 inhibitor therapies for severe COVID-19 disease[48–50].

In addition to the changes in the heterogeneity of neutrophils, a strong decrease in T-cells was observed, especially in subsets that possess cytolytic activity such as CD8, VD1 and VD2 T-cells. These results are consistent with other studies showing a decrease of CD8+ during COVID-19 disease[14,15]. As for VD2 T-cells, which are not MHC-restricted T-cells[51,52], we showed a general decrease in the periphery with disease severity. This is in line with other inflammatory disease such as psoriasis[53] and Crohn's disease[54]. However, in the lungs, during chronic obstructive pulmonary disease, γδ T-cell counts have been reported to be significantly lower in induced sputum (IS) and bronchoalveolar lavage (BAL) but not in peripheral blood, suggesting unclear

inflammatory mechanisms that could influence γδ T-cells counts in the periphery[55]. Interestingly, γδ T-cells, in particular VD2, are known to participate in influenza immune response[56], and actively recruit and activate neutrophils to the site of infection or inflammation[57,58]. Activated, neutrophils have also been found to inhibit γδ T-cells functional capacity, promoting the resolution of inflammation[59,60]. Therefore, it will be essential to investigate the neutrophil to γδ T-cells relashionship present in lungs of SARS-CoV-2 infected patients.

During aging, VD2 T-cell counts in the periphery have been shown to decrease with age for both males and females[61–64]. Interestingly, VD2 counts between males and females can be quite variable depending on the population sampled. Higher VD2 counts in males were observed in a Japanese population, while a similar study in Germany and Italy observed higher VD2 counts in females[61,64]. Additionally, elderly individuals generally have systemic chronic low-grade inflammation, which was previously termed as inflamm-aging[65], with higher basal levels of molecules such as CRP, TNF-a and IL-6[66–68]. These similarities in modulation of VD2 T-cell counts and cytokines between COVID-19 severity and aging could explain why elderly individuals are more susceptible to severe disease, since they have a higher basal level of inflammation and lower level of VD2 T-cells as compared to the young. In any case, the lower VD2 counts in elderly populations will influence the immature neutrophil to VD2 ratio by overestimating their risk to severe COVID-19 as compared to a younger population. However, since age is a very well established risk factor for severe COVID-19 disease[69–71], we postulate that the immature neutrophil to VD2 ratio takes into account the immunological age (measured by VD2 T-cell counts of the patient) which contributes to the improved sensitivity and specificity observed here with area under receiver operating characteristic analysis (Fig. 5).

Our results indicate that an early post illness onset iNVD2R, accessible through a simple 6 colours flow cytometry panel (CD3; VD2; CD66b/CD15; CD16; CD10; CD45), would be an excellent prognostic screening tool for predicting probable patient progression to pneumonia or hypoxia. This prognostic possibility needs to be validated in a prospective cohort. Moreover, CD8 could also be included in the flow cytometry panel as a fallback option since VD2 counts could be decreased by medication, such as Azathioprine, as well as underlying conditions, such as inflammatory bowel disease, aging or psoriasis, which could be risk factors for COVID-19[54]. Analysis of the proposed parameter would allow for a more accurate and earlier prognosis due to the interconnection between neutrophils and Vδ2 T cells, which can then be utilised for early therapeutic interventions, improve patient triage and better healthcare resource management.

## Methods
**Study design**. This was an observational cohort study of patients with PCR-confirmed COVID-19 who were admitted to the National Centre for Infectious Diseases, Singapore. All patients with COVID-19 in Singapore, regardless of the severity of infection, are admitted to isolation facilities until clinical recovery and

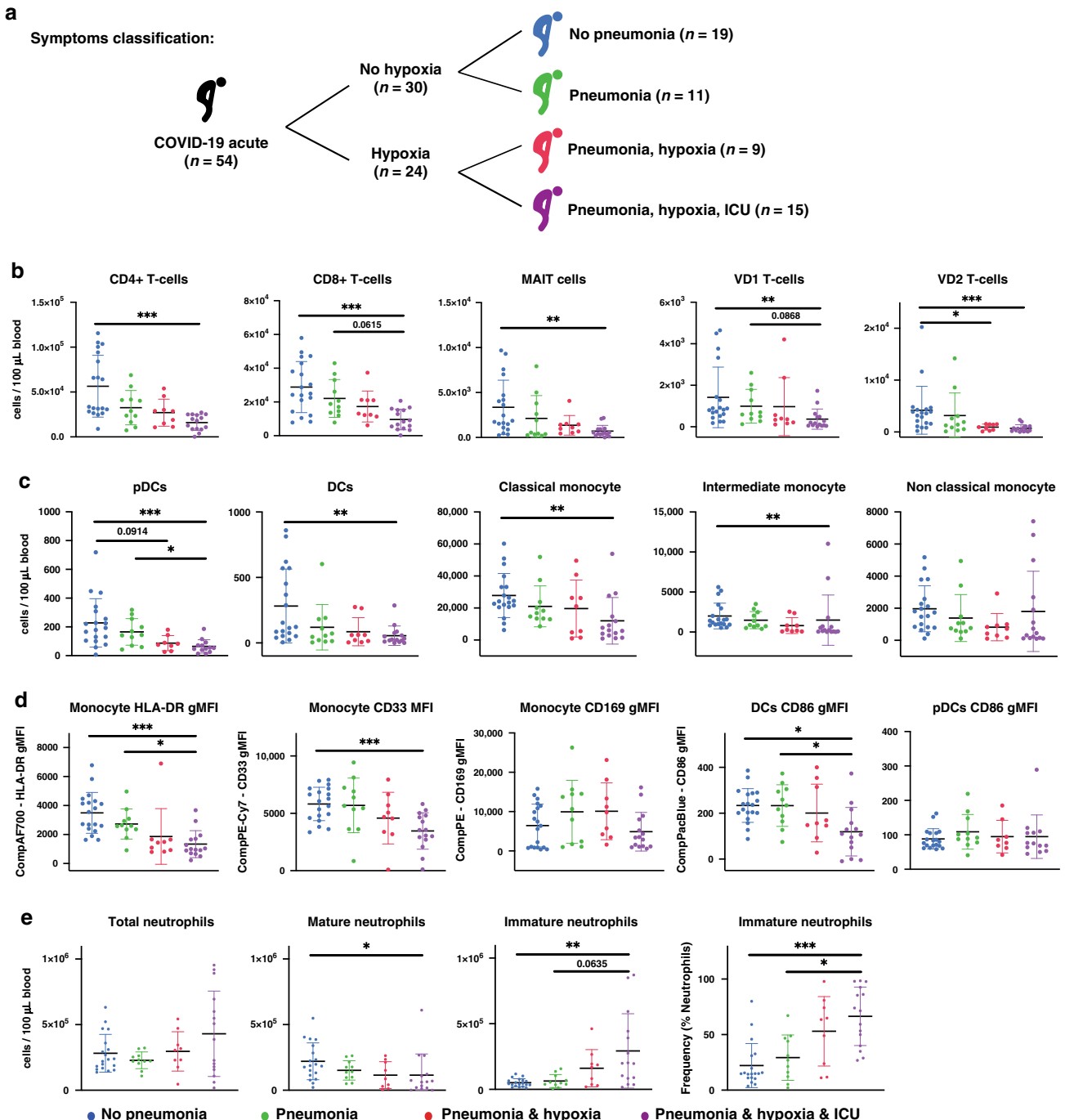

**Fig. 3 Patient symptoms are reflected in immune cell variations. a** Schematic representation of clinical symptoms in the patient cohort. **b** Absolute counts of T-cells across the severity. **c** Absolute counts of antigen presenting cells across the severity. **d** gMFI of activation markers on antigen presenting cells. **e** Absolute counts and frequency in neutrophil compartments. Data presented are from individual human acute COVID-19 patients $n = 54$ from panels **a**, **b** and **c**, separated according to clinical severity: no pneumonia $n = 19$, pneumonia no hypoxia $n = 11$, pneumonia with hypoxia no ICU $n = 9$, pneumonia with hypoxia with ICU $n = 15$. Scatter dot plots are presented with mean ± SD. Absolute counts and frequency were analysed by Kruskal–Wallis with Dunn multiple testing correction, gMFI was analysed by Brown–Forsythe and Welch ANOVA with Dunnett T3 multiple testing correction. *$p < 0.05$, **$p < 0.01$, ***$p < 0.001$. Data available in source data file, exact $p$-values are given in Supplementary Data 1.

viral clearance. Supportive therapy including supplemental oxygen and symptomatic treatment were administered as required. Pneumonia was diagnosed radiologically by interpretation of CXR or CT thorax images. Hypoxia is defined as requirement for supplemental oxygen, which was started if peripheral $O_2$ saturations ($SpO_2$) were <94%. Admission to ICU was reserved for those patients requiring $[FiO_2] \geq 40\%$ or with haemodynamic instability, and included invasive mechanical ventilation when necessary. Incidence of thrombo-embolic and cardiac events are indicated in Supplemental Table 1.

Sample Size: No power analysis was done. Sample size was based on sample availability. Randomisation: No randomisation was done. Blinding: Clinical parameters were made available after data analysis. Patient clinical information was collected using Excel for Mac version 16.16.8 (Microsoft, USA).

**Ethics statement.** Written informed consent was obtained from participants in accordance with the tenets of the Declaration of Helsinki. For COVID-19 blood/

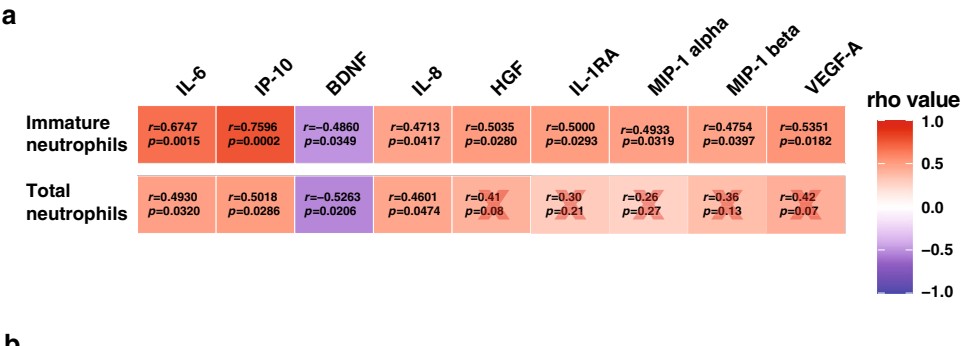

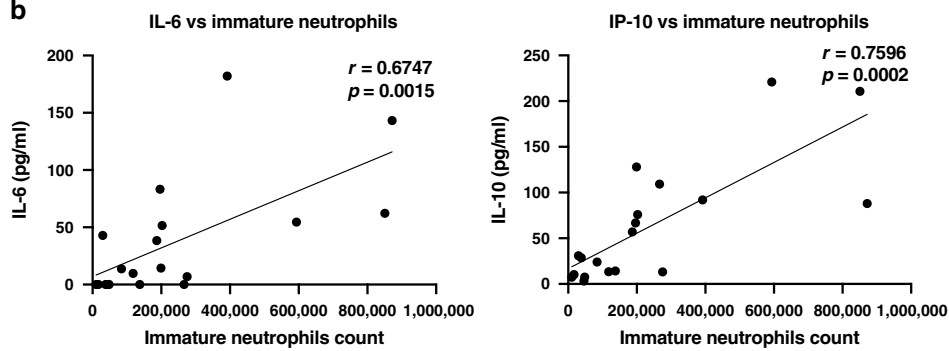

**Fig. 4 Immature neutrophils correlate with several analytes in paired patient plasma. a** Spearman correlations between total neutrophils or immature neutrophils and plasma analytes. Red cross represents non-significant correlations. **b** Individual plots of Spearman correlations between immature neutrophil counts and IL-6 and IP-10. Line was drawn using simple linear regression. Data was analysed using non-parametric Spearman correlation two-tailed function in Prism. Data from $n = 19$ individual acute COVID-19 patients. Data available in source data file.

plasma collection, "A Multi-centred Prospective Study to Detect Novel Pathogens and Characterize Emerging Infections (The PROTECT study group)", a domain specific review board (DSRB) evaluated the study design and protocol, which was approved under study number 2012/00917 the National Healthcare Group (NHG). Healthy volunteers samples were obtained under the following IRB "Study of blood cell subsets and their products in models of infection, inflammation and immune regulation" under the number 2017/2806 by the SingHealth Centralised Institutional Review Board (CIRB).

**Donor information**. Patients who tested PCR-positive for SARS-CoV-2 in a respiratory sample from February to April 2020 were recruited into the study[72]. Demographic data, days post disease onset date (unavailable for 5 asymptomatic patients), clinical score and SARS-CoV-2 RT-PCR results during the hospitalisation period were retrieved from patient clinical records. Relevant information are given in Supplementary Table 1. Patients were classified in different clinical severity groups depending on the presence of pneumonia, hypoxia and the need for ICU hospitalisation. For healthy volunteers, demographic data are provided in Supplementary Table 2. Blood was collected in VACUETTE EDTA tubes (Greiner Bio, #455036) for healthy donors and acute patients, or Cell Preparation Tubes (CPT) (BD, #362761) for recovered patients. 100 μL of whole blood was extracted for each FACS staining panel (Supplementary Table 3).

**Multiplex microbead-based immunoassay**. When available, plasma fraction was harvested after 20 min centrifugation at 1700 x $g$ of blood collected in BD Vacutainer CPT tubes (BD, #362761). Plasma samples were treated by solvent/detergent treatment with a final concentration of 1% (volume) Triton X-100 (Thermo Fisher Scientific, #28314) for virus inactivation at RT for 2 h in the dark under stringent Biosafety laboratory 2+ conditions (approved by Singapore Ministry of Health)[73]. Cytokines detection in Triton-X treatment was compared with untreated samples for healthy donor and was found to be highly correlative for detected cytokines except for sCD40 (Supplemental Fig. 7). Immune mediator levels in COVID-19 patient plasma samples across acute samples were measured with by Luminex using the Cytokine/Chemokine/Growth Factor 45-plex Human ProcartaPlex™ Panel 1 (ThermoFisher Scientific, #EPX450-12171-901). Data acquisition was performed on FLEXMAP® 3D (Luminex) using xPONENT® 4.0 (Luminex) software. Data analysis was done on Bio-Plex Manager™ 6.1.1 (Bio-Rad). Standard curves were generated with a 5-PL (5-parameter logistic) algorithm, reporting values for both mean florescence intensity (MFI) and concentration data. Luminex data was generated from four different runs with each run having a number of samples which are common to the first run. For each subsequent run beyond the first, the mean of the common samples on each of the plates for each analyte was compared to the mean of the same samples in the first run to obtain a correction factor expressed in the following formula:

correction_factor = mean(common_sample_concentration_in_run1)—mean (common_sample_concentration_in_subsequent_run). This correction factor was computed for each plate and analyte combination in the subsequent runs and added to the observed concentration to get the final normalised concentration. In the event that none of the common samples had concentration within the standard curve, no correction was done. Analyte concentrations were logarithmically transformed to ensure normality. Analytes that were not detectable in patient samples were assigned the value of logarithmic transformation of Limit of Quantification (LOQ).

**Flow cytometry**. Whole blood was stained with antibodies as stated in Supplementary Table 3 (100 μL of whole blood per flow cytometry panel) for 20 min in the dark at RT. Samples were then supplemented with 0.5 mL of 1.2× BD FACS lysing solution (BD 349202). Final FACS lysing solution concentration taking into account volume in tube before addition is 1×. Samples were vortexed and incubated for 10 min at RT. 500 μL of PBS (Gibco, #10010-031) was added to wash the samples and centrifugated at 300 $g$ for 5 min. Washing step of samples were repeated with 1 mL of PBS. Samples were then transferred to polystyrene FACS tubes containing 10 μL (10800 beads) of CountBright Absolute Counting Beads (Invitrogen, #36950). Samples were then acquired without delay, with vortexing before and every 3 min during acquisition to minimise fixed cell adherence to the tubes, using BD LSRII 5 laser configuration using automatic compensations and running BD FACS Diva Software version 8.0.1 (build 2014 07 03 11 47), Firmware version 1.14 (BDLSR II), CST version 3.0.1, PLA version 2.0. Analysis of flow cytometric data was performed with FlowJo version 10.6.1. Gating strategies for panels A, B and C are presented in Supplementary Figs. 4–6, respectively. LSRII image in Fig. 1a was created with BioRender.com, and person silhouettes were modified from public domain art (clipartkey.com).

**Statistical analysis**. Statistical analysis was performed using Prism 7.03 to 8.3.0 (Graph Pad Software, Inc). For comparisons of absolute cell counts or frequency, Kruskal-Wallis Test corrected with Dunn's method was performed. For comparisons of geometric Mean Fluorescence Intensity (gMFI) between three or more independent groups, Brown-Forsythe and Welch ANOVA using Dunnett T3 correction for multiple comparison was performed. For comparisons of geometric Mean Fluorescence Intensity (gMFI) between two independent groups, Brown-Forsythe and Welch ANOVA without correction for multiple comparison was performed. For correlation analysis, spearman rank correlation was performed. $p$-Values < 0.05 for correlations, while adjusted $p$-values <0.05 for all the other comparisons were considered significant. Exact p-values are available in Supplementary Data 1.

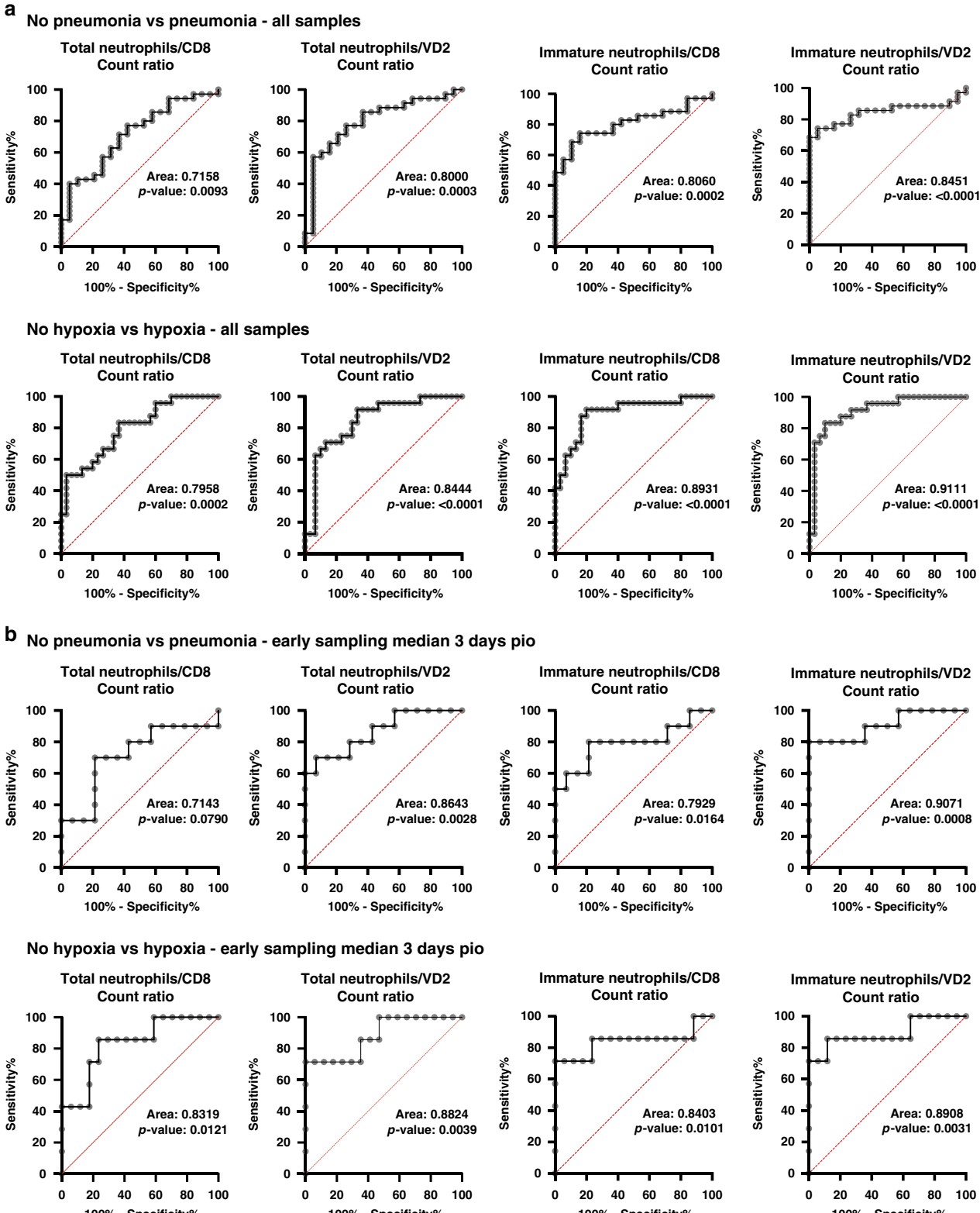

**Fig. 5 Immature neutrophil to VD2 T-cell ratio is an early prognosis marker for pneumonia and hypoxia symptoms. a** ROC curve analysis comparison was performed for pneumonia and hypoxia symptoms between absolute counts of total neutrophils to CD8 T-cell ratio, total neutrophils to VD2 T-cell, immature neutrophils to CD8 T-cell ratio, and immature neutrophils to VD2 T-cell ratio, $n = 54$ individual acute COVID-19 patient samples. **b** Similar analysis was performed on a subset of early samples from the 54 acute patients ($n = 24$ individual acute COVID-19 patients following the criteria: sampled at 1 to 7 days pio. Median of this subset is 3 days pio). ROC curve was analysed using Wilson/Brown method, 95% confidence interval and standard error for panel A are given in Supplementary Data 1 and for panel B are given in Table 1. Data available in source data file.

**Table 1 ROC curve analysis for neutrophils to T-cell ratios in patients with pneumonia or hypoxia compared to those without as presented in Fig. 5b.**

| Variable | Pneumonia | | | Hypoxia | | |
|---|---|---|---|---|---|---|
| | AUC (95% CI) | Std. error | p-Value | AUC (95% CI) | Std. error | p-Value |
| Total neutrophils/CD8 T-cells | 0.7143 (0.4909-0.9377) | 0.1140 | 0.0790 | 0.8319 (0.6526-1) | 0.09149 | 0.0121 |
| Total neutrophils/VD2 T-cells | 0.8643 (0.7135-1) | 0.07694 | 0.0028 | 0.8824 (0.7239-1) | 0.08083 | 0.0039 |
| Immature neutrophils/CD8 T-cells | 0.7929 (0.5884-0.9973) | 0.1043 | 0.0164 | 0.8403 (0.6079-1) | 0.1186 | 0.0101 |
| Immature neutrophils/VD2 T-cells | 0.9071 (0.7754-1) | 0.06723 | 0.0008 | 0.8908 (0.7160-1) | 0.08915 | 0.0031 |

ROC analysis was performed on COVID-19 patients between 2 to 7 days pio (24 patients, median 3 days pio). ROC curve was built by plotting true positive rate (sensitivity) against false positive rate (100%- sensitivity) and AUC was calculated from the plot using the Wilson/Brown method. ROC receiver operating characteristic, AUC area under curve, CI confidence interval, Std. error standard error.

**Data analysis and UMAP visualisation**. UMAP: Gated cells were manually exported using FlowJo (Tree Star Inc.). Samples were then used for UMAP analysis using cytofkit2 R Packages with RStudio v3.5.2[74]. Five healthy, six acute and four recovered patients were each concatenated to its respective groups and 100000 cells were analysed using the ceil method. Custom R scripts were used to generate Z-score and correlation heatmaps.

**Reporting summary**. Further information on research design is available in the Nature Research Reporting Summary linked to this article.

## Data availability

Custom R scripts are available in Supplementary Software 1. Other data not made publicly available due to protection of patients' confidentiality can be obtained upon request to the corresponding author. Source data are provided with this paper.

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

## Acknowledgements

Authors would like to acknowledge all the support received on this project from the Singapore Immunolgy Network (SIgN), LN lab members Chek Meng Poh and Anthony Torres Ruesta, and SIgN flow cytometry facility, especially Ivy Chay Huang Low. We would also like to thank the study participants who donated their blood samples to this project, and the healthcare workers caring for COVID-19 patients. This work was supported by Singapore Immunology Network core research grant, the A*STAR COVID-19 Research funding (H/20/04/g1/006) provided to Singapore Immunology Network by the Biomedical Research Council (BMRC), A*STAR. Subject recruitment and sample collection were funded by the National Medical Research Council (NMRC) COVID-19 Research fund (COVID19RF-001). The SIgN flow cytometry and the Multiple analyte platforms were supported a grant from the National Research Foundation, Immunomonitoring Service Platform ISP) (#NRF2017_SISFP09).

## Author contributions

G.C., W.X. and I.K. conceptualised, designed the panels, acquired, analysed and interpreted the data, and wrote the manuscript. M.Y.A. processed the patient blood, stained and fixed the samples. Y.H.C., S.W.F., K.J.P., C.Y.P.L., N.K.W.Y., S.N.A., R.S.L.C., W.H., A.A. and B.L. acquired and analysed the data. S.S.W.C., B.E.F., B.E.Y., Y.S.L. and D.C.L. designed and supervised sample collection. O.R., L.R., L.G.N., A.L. and L.F.P.N. conceptualised, designed, analysed and wrote the manuscript. All authors revised and approved the final version of the manuscript.

## Competing interests

The authors declare no competing interests.
