## [Peer Review File · Nature Communications]

Reviewers' Comments:

Reviewer #1:

Remarks to the Author:

COVID-19 disease is known to be associated with significant immune cell disbalance and strong induction of inflammatory cytokines, resembling in some instances a cytokine storm. In this study, the authors set out to perform an in-depth immunophenotyping of a large number of immune cell markers in whole blood samples from 54 COVID-19 patients with different clinical conditions and in 19 healthy control subjects. The innovative aspect of the study is the analysis of whole blood (i.e., total leukocytes including the major population of granulocytes) as compared to Ficoll-separated PBMC (i.e. mononuclear cells lacking most of granulocytes), which has been used in most previous studies.

The results showed a decline in several but not all of the analyzed leukocyte subsets. When comparing acute COVID-19 versus recovered COVID-19 patients and healthy controls, the most conspicuous observations were the expression of a number of activation markers on monocytes and neutrophils, and more specifically, an increase of immature neutrophils (CD16^{low}/highCD10⁻) in acute COVID-19 patients. Furthermore, there was a significant decline of CD8 and Vdelta2 (Vd2) T cells. Further stratification of the patients according to the clinical condition revealed a gradual reduction of certain T-cell populations (specifically CD8 and Vd2) with more severe disease, and a concomitant increase of immature neutrophils despite the decrease of mature neutrophils. Based on these results the authors calculated an immature neutrophils-to-Vd2 index which correlated well with disease activity (pneumonia yes/no; hypoxia yes/no). The predictive value of the index was retrospectively verified in samples of patients which had been collected very early after onset of illness.

Overall, the multiparameter FACS analysis has been carefully conducted, and results reveal the importance of analyzing whole blood samples. The statistical analysis is appropriate for this kind of data. The gating strategy for the three panels of FACS antibodies as shown in Supplementary Figures is appropriate. The predictive value of the newly established cell index will need to be confirmed in larger studies.

Specific remarks:

It is unclear whether the group of "recovered samples" (n=28) is included in the group of "acute samples" (n=54) or not (Supplementary Table 1). If so, then it should be feasible to follow up individual patients (i.e., show how parameters change in individual patients from "acute" to "recovered" over time).

Reviewer #2:

Remarks to the Author:

Carissimo et al. have analyzed blood samples from 54 COVID-19 patients by using 3 different panels of mAbs, detected by flow cytometry. Authors finely describe some main phenotypic characteristics of monocytes, granulocytes and T cells, including CD4⁺, CD8⁺, MAIT, and gamma delta cells. They found that correlations exist between the severity of the disease and several parameters linked to innate immunity, including the number of immature neutrophils.

Comments.

1. I realize that performing a comprehensive study like this in the moment of pandemic was extremely

difficult, and I appreciate all the efforts done and the data presented here. However, it would be extremely interesting to see functional data on some of the cells here described, particularly immature neutrophils. Do authors have any data on cytokine production by these cells, or by monocytes, possibly in patients with different forms of COVID-19?

2. Lane 70, correct "studies" with studies

3. Figure 3A: symptom classification can be explained in the text or in the figure legend, and pane A could be removed

4. Figure 2C: please specify in the legend that this is a representative example of CD8+ T cells, and quote the gating strategy to arrive here

5. Reference 13: correct "Biasi" in "De Biasi"

6. Lane 278 and Reference 52: the concept of "inflammaging" was not termed by the authors, nor does it come from this reference, published in 2017 by one author of this paper. The term derives from the group of Claudio Franceschi, about 20 years ago (see: *Ann NY Acad Sci.* 2000 Jun;908:244-54).

7. Do Authors have data concerning plasma levels of other cytokines related to innate immunity, since a 45-plex was used?

8. References: there are some citations of uploaded but non-published manuscripts. If some of them have been published, please update.

Reviewer #3:

Remarks to the Author:

Carissimo et al present a descriptive study on leukocyte numbers in the blood of COVID-19 patients. They tested the hypothesis that hyper-inflammation plays an important role in COVID19 and that there is a correlation between disease severity and extent of such inflammation. They compared different disease severities with each other and compared the COVID data with those of healthy controls. They conclude on the basis of many ROC curves that the ratio immature neutrophils/VD2 T-cells is (the best) prognostic marker for deterioration of disease. The article would benefit when the following issues are carefully addressed.

Major (random order):

1. The pathogenesis and main cause of death in COVID-19 are associated with aberrant coagulation and (lung) tissue edema. The majority of critical cases are characterized by thrombo-embolic complications in the lung vasculature. Here ACE2 plays an important role and bradykinin metabolism seems critically involved in the disease mechanism. This of course does not rule out that inflammation takes part in the disease process but it should be carefully placed in the total pathogenesis of COVID-19 particularly in an article in a general science journal such as *Nature Communications*.

2. Most of the data in the article are not COVID-19 specific, but generally seen in critical illness. In fact, the data fit with the well-known left-shift in neutrophils in all kind of severe diseases. The data really need benchmarking with other patients with other systemic inflammatory conditions. So more left shift (more progenitors, banded cells) in more severe disease is something to be expected. If this is not possible indicate the reason why and at least discuss this possibility in the discussion section.

3. The most important figure seems to be figure 5B. However, this is not easy to understand and

needs more information.

- For a marker to be predictive, ROC analysis is the first step but is not sufficient. The next step is to show in a prospective study in new population of patients that the marker is predictive. It is clear that this is impossible now as the number of patients is dropping. So please refrain from using the term predictive in the title of the article.

- The authors use a 'median 3 days pio' in their figure and mention an median admittance to the hospital 7 days pio. This really needs explanation how this is done. It is highly likely that there is a selection bias towards patients with an early pio sample available. What is the spread in pio of the two groups? The different times between disease onset and analysis is not appropriate. The need for a prospective cohort is now even more imminent.

4. There seem important inconsistencies in the article that need attention and explanation:

- healthy donors are not age matched and this should be carefully addressed.

- Figure 1: it should really be explained why the recovered patients (time of recovery is 30 days) do not normalize their neutrophil numbers? Are these paired samples? If not, please provide detailed demographics. Otherwise, these data are difficult to interpret.

- Figure 1D/E: the use of a relative Z-score in pseudo-colors for single neutrophil activation markers that is difficult to interpret in terms quantification of activation.

- Figure 1G: this 'representative' figure suggests that almost all neutrophils are CD10- in acute COVID-19 patients. Figure 1H clearly shows that this is not the case. Please avoid such confusion.

- It is not clear whether mean or median is depicted in Figure F and H. It seems like the mean is depicted whereas the data is not normally distributed

- The data of 1F and 1H imply a 'normal' left shift often seen in severe disease.

- Figure 2C see comment 1D/E.

5. Also figure 3E implies a 'normal' left shift. So the data of figures 1D/E and 3E basically need a comparison with other severe (inflammatory?) PCR-negative disease samples(see comment 1).

6. Figure 1D and 4B imply that in COVID-19 up to 90% of the neutrophils are immature in very high numbers. The data in fact show a low expression of CD10. How do the authors know the cells are immature other than low CD10 expression? Hidalgo et al (Trends in Immunology 2019) published a strategy on identifying progenitors on the basis of expression of CD11b and CD16 that was supported by cytopins. Therefore, supportive data of cytopins will show whether the cells have immature characteristics: banded or round nuclei, dark cytoplasm etc. or that something is else is happening

7. In Figure 2D the CD38 expression of total CD4 and CD8 is not very useful as it is shown in Figure 2C that the composition of the CD4 and CD8 T cell pool is very different in patients and in Fig 2E that CD38 is very different between the subsets even in healthy controls.

8. In Figure 2E it seems odd that CM, EM and TEMRA have lower CD38 gMFI than naïve for both CD4 and CD8 in healthy controls. Also, can the CD38 be reliably measured in CD4 TEMRA while they are almost absent (Sup Fig 2)?

Minor:

9. Please add the defining clinical characteristics for pneumonia and hypoxia. For hypoxia it seems that an FiO₂ of 40% is used, which is not necessarily an indication for admittance of the ICU at least not in other countries. Also add other characteristics of the patients to this description: particularly the presence of thrombo-embolic events.

10. The presence of neutrophils in samples obtained by bronchoscopy is important (ref. 26). Unfortunately, this is based on a not (yet) peer-reviewed study and this should be indicated as such.

11. Line 329: it is of great concern that two different blood collection tubes were used with different anti-coagulants: EDTA and NaHep. It seems that cytokines were measured in EDTA plasma and cells were analyzed in NaHep. This should be clear and conclusions on the correlations of cytokines and neutrophils in the peripheral blood should be more carefully addressed.

12. Line 336: please provide the data that solvent/detergent treatment does not affect the measured cytokines or show the difference in the online supplement.

13. Line 348: please provide the calculations on which the correction factor is obtained. These can be added to the online supplement.
14. FACS analysis in PBS can be problematic for neutrophils and monocytes as they avidly bind to bare plastic surfaces. Most people use PBS with added proteins (e.g. albumin) to minimize this behavior. Please indicate that PBS alone did not lead to lower cell counts in the flowcytometer.
15. Please indicate that the patients were not having co-infections with bacteria.

REVIEWER COMMENTS

Reviewer #1 (Remarks to the Author):

COVID-19 disease is known to be associated with significant immune cell disbalance and strong induction of inflammatory cytokines, resembling in some instances a cytokine storm. In this study, the authors set out to perform an in-depth immunophenotyping of a large number of immune cell markers in whole blood samples from 54 COVID-19 patients with different clinical conditions and in 19 healthy control subjects. The innovative aspect of the study is the analysis of whole blood (i.e., total leukocytes including the major population of granulocytes) as compared to Ficoll-separated PBMC (i.e. mononuclear cells lacking most of granulocytes), which has been used in most previous studies.

The results showed a decline in several but not all of the analyzed leukocyte subsets. When comparing acute COVID-19 versus recovered COVID-19 patients and healthy controls, the most conspicuous observations were the expression of a number of activation markers on monocytes and neutrophils, and more specifically, an increase of immature neutrophils (CD16^{low}/highCD10⁻) in acute COVID-19 patients. Furthermore, there was a significant decline of CD8 and Vdelta2 (Vd2) T cells. Further stratification of the patients according to the clinical condition revealed a gradual reduction of certain T-cell populations (specifically CD8 and Vd2) with more severe disease, and a concomitant increase of immature neutrophils despite the decrease of mature neutrophils. Based on these results the authors calculated an immature neutrophils-to-Vd2 index which correlated well with disease activity (pneumonia yes/no; hypoxia yes/no). The predictive value of the index was retrospectively verified in samples of patients which had been collected very early after onset of illness.

Overall, the multiparameter FACS analysis has been carefully conducted, and results reveal the importance of analyzing whole blood samples. The statistical analysis is appropriate for this kind of data. The gating strategy for the three panels of FACS antibodies as shown in Supplementary Figures is appropriate. The predictive value of the newly established cell index will need to be confirmed in larger studies.

Specific remarks:

It is unclear whether the group of “recovered samples” (n=28) is included in the group of “acute samples” (n=54) or not (Supplementary Table 1). If so, then it should be feasible to follow up individual patients (i.e., show how parameters change in individual patients from “acute” to “recovered” over time).

Answer: We thank the reviewer for the positive comments. We agree with the reviewer that it would have been ideal to be able to collect paired recovered samples in order to allow individual patient follow up. Unfortunately, we only managed to collect paired-recovered samples for 11 patients and this explains why we favoured increasing the data points rather than performing paired analysis. We have now included the information about paired samples in the recovered group in the Supplemental Table 1, and also indicated this point in the text on lines 103-104, page 6 to read: “Unfortunately, only 11 patients had paired acquisition between acute and recovered which prevented meaningful paired analysis (Supplementary Table 1).”

Reviewer #2 (Remarks to the Author):

Carissimo et al. have analyzed blood samples from 54 COVID-19 patients by using 3 different panels of mAbs, detected by flow cytometry. Authors finely describe some main phenotypic characteristics of monocytes, granulocytes and T cells, including CD4+, CD8+, MAIT, and gamma delta cells. They found that correlations exist between the severity of the disease and several parameters linked to innate immunity, including the number of immature neutrophils.

Comments.

1. I realize that performing a comprehensive study like this in the moment of pandemic was extremely difficult, and I appreciate all the efforts done and the data presented here. However, it would be extremely interesting to see functional data on some of the cells here described, particularly immature neutrophils. Do authors have any data on cytokine production by these cells, or by monocytes, possibly in patients with different forms of COVID-19?

Answer: We agree with the reviewer that functional data on these cells would be very informative. Unfortunately, the strict biosafety regulations in Singapore do not allow manipulation of SARS-CoV-2-infected live cells of acute patients. The direct antibody staining in whole blood followed by fixation procedure was the only approved workflow.

For functional data of these cells, we strongly suspect that the cells described in this preprint (<https://www.medrxiv.org/content/10.1101/2020.05.22.20106724v1.full>) are in fact immature neutrophils. Authors describe “*CD16^{int}CD44^{low}CD11b^{int} low-density inflammatory band (LDIB) neutrophil population that trends over time with changes in disease status. These cells demonstrated spontaneous neutrophil extracellular trap (NET) formation, phagocytic capacity, enhanced cytokine production, and associated clinically with D-dimer and systemic IL-6 and TNF- α levels, particularly for CD40⁺ LDIBs.*”

We have now included this information and cited this preprint in the revised manuscript discussion on lines 252 to 257, page 12 to read: “Consistent with our data, a recent non peer reviewed study on a small number of patients reported that the presence of “low density inflammatory neutrophils” was strongly associated with disease severity and IL-6 levels³³. Functionally these low density neutrophils showed spontaneous extracellular trap formation, enhanced cytokine production and associated with D-dimer and systemic IL-6 and TNF- α levels³³.”

2. Lane 70, correct "stuiies" with studies

Answer: We have corrected the typing error.

3. Figure 3A: symptom classification can be explained in the text or in the figure legend, and pane A could be removed

Answer: Indeed, the illustration could be removed and replaced by an in-text explanation. However, we feel that the illustration will allow better understanding of the classification and the different colour codes used in the figure. In the revised manuscript, explanation of the classification reads on lines 172 to 174 pages 9: “In order to associate the data with the

clinical severity we separated the patients into four different groups: no pneumonia, pneumonia only, pneumonia and hypoxia, and pneumonia and hypoxia requiring ICU admission (Figure 3A)".

4. Figure 2C: please specify in the legend that this is a representative example of CD8+ T cells, and quote the gating strategy to arrive here

Answer: The legend has been modified accordingly to read: "left panel: CD45RA and CD27 gating strategy example on CD8+ T-cells".

5. Reference 13: correct "Biasi" in "De Biasi"

Answer: This preprint reference was updated to the recently published (in Nat Comms) version with the correct first author name as De Biasi.

6. Lane 278 and Reference 52: the concept of "inflammaging" was not termed by the authors, nor does it come from this reference, published in 2017 by one author of this paper. The term derives from the group of Claudio Franceschi, about 20 years ago (see: Ann NY Acad Sci. 2000 Jun;908:244-54).

Answer: We apologize the oversight and agree that the "we" was misleading. It has now been replaced with "was" and is followed by the original 2000 reference. We have also added the following reference from a 2020 short review on inflamm-aging and SARS-CoV-2.

Bonafe, M., Prattichizzo, F., Giuliani, A., Storci, G., Sabbatinelli, J. & Olivieri, F. Inflamm-aging: Why older men are the most susceptible to SARS-CoV-2 complicated outcomes. *Cytokine Growth Factor Rev* **53**, 33-37, doi:10.1016/j.cytogfr.2020.04.005 (2020).

The discussion on lines 302 to 305 now reads: "During aging, CD2 T-cell counts in the periphery have been shown to decrease with age. Elderly individuals generally have systemic chronic low-grade inflammation, which was previously termed "inflamm-aging"⁶⁰, with higher basal levels of molecules such as CRP, TNF- α and IL-6⁶¹⁻⁶³."

7. Do Authors have data concerning plasma levels of other cytokines related to innate immunity, since a 45-plex was used?

Answer: Multiplexed luminex assays have the pitfall of lower sensitivity. We have presented the analysed correlations for cytokines that were consistently detected above the lower detection limit across the patients. The 45 analytes concentrations are available in the source data file.

8. References: there are some citations of uploaded but non-published manuscripts. If some of them have been published, please update.

Answer: Yes, we have reconfirmed all cited preprints for the peer-reviewed published versions at each submission stage and we will continue to update during the proofs stage (hopefully).

Reviewer #3 (Remarks to the Author):

Carissimo et al present a descriptive study on leukocyte numbers in the blood of COVID-19 patients. They tested the hypothesis that hyper-inflammation plays an important role in COVID19 and that there is a correlation between disease severity and extend of such inflammation. They compared different disease severities with each other and compared the COVID data with those of healthy controls. They conclude on the basis of many ROC curves that the ratio immature neutrophils/VD2 T-cells is (the best) prognostic marker for deterioration of disease. The article would benefit when the following issues are carefully addressed.

Major (random order):

1. The pathogenesis and main cause of death in COVID-19 are associated with aberrant coagulation and (lung) tissue edema. The majority of critical cases are characterized by thrombo-embolic complications in the lung vasculature. Here ACE2 plays an important role and bradykinin metabolism seems critically involved in the disease mechanism. This of course does not rule out that inflammation takes part in the disease process but it should be carefully placed in the total pathogenesis of COVID-19 particularly in an article in a general science journal such as Nature Communications.

Answer: We are grateful for the suggestions to place the manuscript in a broader context. We have now provided more context on coagulopathy in the introduction on lines 65-72, page 4 to read:

“The majority of critical cases of COVID-19 are associated with coagulopathy with a high prevalence of thromboembolic events in patients under mechanical ventilation which lead to inclusion of anticoagulation therapies in the standard of care of severe COVID-19 cases ⁶⁻⁸. While the strong inflammatory response to COVID-19 has been proposed to be associated to COVID-19 associated coagulopathy ⁶, it remains unclear how SARS-CoV-2 infection affects the activation of immune cells and their contribution towards the different severity of disease outcomes in patients.”

However, we feel that discussing the virus entry receptor ACE-2 or the potential role of bradykinin metabolism downstream of viral reduction of ACE-2 would be more appropriate in the context of a review article and out of the scope of this study.

2. Most of the data in the article are not COVID-19 specific, but generally seen in critical illness. In fact, the data fit with the well-known left-shift in neutrophils in all kind of severe diseases. The data really need bench marking with other patients with other systemic inflammatory conditions. So more left shift (more progenitors, banded cells) in more severe disease is something to be expected. If this is not possible indicate the reason why and at least discuss this possibility in the discussion section.

Answer: We wish to stress that the identification of high immature neutrophil counts in our study specifically showed that in COVID-19 patients there is an accumulation of immature forms of neutrophils which is consistent with other studies. Furthermore, we show that the neutrophil-to-lymphocyte ratio (NLR) proposed by various groups to predict disease outcome in COVID-19 could be improved by looking at immature counts and VD2 T-cells (or CD8)

instead to increase the sensitivity and accuracy of this severity indicator. Unfortunately, we do not have access to other cohorts to benchmark this possibility during the current pandemic. Nonetheless, we note the reviewer's point and have discussed this limitation in the revised manuscript on lines 259 to 267, page 12-13 to read:

"Interestingly, the diagnostic value of a neutrophil "left shift" (banded versus segmented neutrophils) had previously been explored in order to predict infectious diseases in addition to inflammatory diseases³⁴ and is therefore not limited to COVID-19 severity. Similarly, the presence of immature low density neutrophils have been reported in the literature for various infectious and inflammatory diseases³⁵⁻³⁷ as well as induced by LPS injection in healthy subjects³⁸, highlighting the necessity of future studies to compare the role and function of these COVID-19 immature neutrophils with the circulating immature neutrophils present in other diseases."

3. The most important figure seems to be figure 5B. However, this is not easy to understand and needs more information.

- For a marker to be predictive, ROC analysis is the first step but is not sufficient. The next step is to show in a prospective study in new population of patients that the marker is predictive. It is clear that this is impossible now as the number of patients is dropping. So please refrain from using the term predictive in the title of the article.

Answer: We have removed the term prognostic from the title. In addition, we have included the following statement in the discussion: "This prognostic possibility needs to be validated in a prospective cohort." on lines 314 to 315, page 15.

- The authors use a 'median 3 days pio' in their figure and mention an median admittance to the hospital 7 days pio. This really needs explanation how this is done. It is highly likely that there is a selection bias towards patients with an early pio sample available. What is the spread in pio of the two groups? The different times between disease onset and analysis is not appropriate. The need for a prospective cohort is now even more imminent.

Answer: We wish to stress that the days pio refers to the days post illness onset (onset self-reported by the patients) at which the FACS sample was acquired. Patient symptoms information and days pio were made available after flow cytometry analysis. The group that is referred to as median 7 days pio is the full acute patient dataset. Median pio was calculated by doing the median of all datapoints for the acute patients (median and inter quartile range now included in the supplementary table 1).

The group at median 3 days pio (Figure 5B) is part of the previous group that fits the following criteria (days pio at time of FACS acquisition of 7 days pio or lower, median was calculated similarly). This is now rephrased for clarity in the manuscript on lines 229 to 232, page 11 to read: "To assess if this analysis could have predictive prognostic value in hospitalisation settings to improve patient management, we repeated the same analysis with only samples that were acquired at or before 7 days pio amongst the 54 acute patients (24 patients, median pio = 3 days, range 1 to 7 days pio)".

We have also modified the Figure 5 B legends to read: "Similar analysis was performed on a subset of early samples from the 54 acute patients (24 patients, 1 to 7 days pio with a median of 3 days pio)."

In addition, we have now added the following statement in the discussion on lines 314-315, page 15: “This prognostic possibility needs to be validated in a prospective cohort.”

4. There seem important inconsistencies in the article that need attention and explanation:
- healthy donors are not age matched and this should be carefully addressed.

Answer: We wish to clarify that there are no inconsistencies, but the fact that the healthy donors are not aged matched is a limitation. We wish to stress that collection of controls was only possible through internal recruitment during the lockdown period in Singapore which prevented us from collecting aged/sex-matched controls. We now discuss this limitation on lines 309 to 310, page 14 to read: “Therefore, it is important to interpret our results between healthy donors and acute samples with the age and sex bias in mind (Supplementary Tables 1 and 2).”.

- Figure 1: it should really be explained why the recovered patients (time of recovery is 30 days) do not normalize their neutrophil numbers? Are these paired samples? If not, please provide detailed demographics. Otherwise, these data are difficult to interpret.

Answer: We wish to reiterate that, as mentioned in the manuscript, these are not paired samples, detailed demographics are provided in Supplementary Table 1. We are unsure as to what the reviewer is referring to, the recovered patient samples showed similar neutrophil counts as healthy donors since no statistical significance was found between healthy donors and recovered patients in total neutrophils counts (Figure 1F) or in immature and mature counts (Fig1H). Of note, for recovered patients the median pio is 30 days, but the range is 11 to 50 days pio with and IQR of 22.5-35.8 days pio (information now added in Supplementary Table 1).

- Figure 1D/E: the use of a relative Z-score in pseudo-colors for single neutrophil activation markers that is difficult to interpret in terms quantification of activation.

Answer: We have now replaced this representation with an interleaved box plot showing all data points for acute versus healthy donors. For ease of reviewing, the new Figures 1D and E are appended here:

Figure 1: SARS-CoV-2 infection induces a decrease in immune cells in peripheral blood. (e) Neutrophil activation markers mean geometric MFI (gMFI) represented as interleaved min to max box plot with all data points shown. (f) Absolute neutrophil counts. (g) Representative plot of mature and immature neutrophil gating strategy in healthy control or acute COVID-19 patient.

- Figure 1G: this 'representative' figure suggests that almost all neutrophils are CD10⁻ in acute COVID-19 patients. Figure 1H clearly shows that this is not the case. Please avoid such confusion.

Answer: We wish to clarify that the aim of this panel was to illustrate an ideal gating of mature and immature neutrophils using the CD16 and CD10 markers (after CD3⁻ CCR3⁻ CD49d⁻ CD11b⁺ CD66b⁺ gating). We now provide a different flow cytometry plot for Fig1G which has a mix of mature and immature populations, see the new Figure 1G panel below:

Figure 1: SARS-CoV-2 infection induces a decrease in immune cells in peripheral blood. (g) Representative plot of mature and immature neutrophil gating strategy in healthy control or acute COVID-19 patient.

- It is not clear whether mean or median is depicted in Figure F and H. It seems like the mean is depicted whereas the data is not normally distributed

Answer: All individual plots are represented as scatter dot plots with mean \pm SD. This is now clearly indicated in each figure legend that contains scatter dot plots.

- The data of 1F and 1H imply a 'normal' left shift often seen in severe disease.

Answer: Yes, the data indeed implies a shift from mature neutrophils to immature neutrophils.

- Figure 2C see comment 1D/E.

Answer: In this particular instance, we believe that the heatmap representation allows for a synthetic representation of the changes in frequency instead of the 16 individual plots representation shown in Supplementary Figure 2. We do not believe that moving the 16 plots to Figure 2 will improve the representation.

5. Also figure 3E implies a 'normal' left shift. So the data of figures 1D/E and 3E basically need a comparison with other severe (inflammatory?) PCR-negative disease samples (see comment 1).

Answer: Indeed, this possibility and interesting comparisons are extensively discussed in the manuscript (see previous responses to comment 2). Unfortunately, we do not have access to such cohorts in the current pandemic context.

6. Figure 1D and 4B imply that in COVID-19 up to 90% of the neutrophils are immature in very high numbers. The data in fact show a low expression of CD10. How do the authors know the cells are immature other than low CD10 expression? Hidalgo et al (Trends in Immunology 2019) published a strategy on identifying progenitors on the basis of expression of CD11b and CD16 that was supported by cytopins. Therefore, supportive data of cytopins will show whether the cells have immature characteristics: banded or round nuclei, dark cytoplasm etc. or that something is else is happening

Answer: We understand the reviewer's interrogation. Indeed, in the review article of Hidalgo et al., 2019, the gating strategy for immature subsets of neutrophils relies on CD16 and CD11b expression to identify neutrophil progenitors and precursors in human bone marrow samples. This strategy specifically requires CD62L and CD16 to discriminate between immature banded neutrophils (CD62L+CD16+) and mature poly-segmented neutrophils (CD16++CD62L+) in healthy donor bone marrow or cord blood. However, it is known that CD62L shedding occurs during neutrophil activation which makes the identification of immature neutrophils difficult during inflammation. Moreover, CD16 also upregulates upon activation (See Fig1E of revised manuscript) which would make identifying immature neutrophils even more difficult. Immature neutrophils can be more easily discriminated using other markers such as CD10, which is used in our gating strategy and is in agreement with the review article mentioned by the reviewer.

The portion of this review article by Hidalgo et al., 2019 that the reviewer is referring to, reads: "Flow sorting of the different populations and subsequent analysis of the resulting cytopsin preparations demonstrates that it is possible to identify and isolate the different maturing forms of neutrophils in the bone marrow and peripheral blood. Additional markers,

such as CD10, CD13, CD64, and CD87, can be used to facilitate the discrimination between mature and immature neutrophils [45–47].“

References 45 to 47 are as follow:

45. Marini, O. et al. (2017) Mature CD10+ and immature CD10– neutrophils present in G-CSF-treated donors display opposite effects on T cells. *Blood* 129, 1343–1356
46. Ng, L.G. et al. (2019) Heterogeneity of neutrophils. *Nat. Rev. Immunol.* 255–265
47. Elghetany, M.T. et al. (2004) Flow cytometric study of neutrophilic granulopoiesis in normal bone marrow using an expanded panel of antibodies: correlation with morphologic assessments. *J. Clin. Lab. Anal.* 18, 36–41”

Specifically, in the study by Marini and colleagues, this morphological confirmation of CD10- and CD10+ neutrophils have been shown with cytospin analyses. The study specifically showed that CD16+CD10- neutrophils contained banded immature phenotypes while CD16+CD10+ neutrophils solely contained mature poly-segmented nuclei phenotypes. Please note that our panel uses a CD16 and CD10 gate after a gate on CD11b and CD66b as well as some exclusion markers. Nevertheless, we had hoped to be able to perform morphology analysis, but the Singapore Ministry of Health Biosafety regulatory board did not allow any workflow to perform blood smears or cytospin analyses on SARS-CoV-2 acute patient samples.

7. In Figure 2D the CD38 expression of total CD4 and CD8 is not very useful as it is shown in Figure 2C that the composition of the CD4 and CD8 T cell pool is very different in patients and in Fig 2E that CD38 is very different between the subsets even in healthy controls.

Answer: The reviewer is correct, and we have now removed the illustrative panel 2D from Figure 2 since it had redundant information with the other panels.

8. In Figure 2E it seems odd that CM, EM and TEMRA have lower CD38 gMFI than naïve for both CD4 and CD8 in healthy controls. Also, can the CD38 be reliably measured in CD4 TEMRA while they are almost absent (Sup Fig 2)?

Answer: We agree with this comment, the frequencies and counts of CD4 TEMRA are very low, which renders the measurement of CD38 probably inconsistent. We have now removed this population from the figure and indicated why in the legends. We have removed mention of this population in the manuscript text.

However, for the CD38 expression in other T-cell populations, our results are consistent with several studies indicating that basal CD38 expression is higher in naïve T-cell populations. After differentiation, basal CD38 expression is lower than in naïve T-cells but will re-express CD38 when activated. For this reason, all comparisons were made using healthy donor CD38 gMFI as baseline within the specific CD4, CD8, VD1, VD2 and MAIT T-cell compartments. Please see below the extracted figures from two studies on human T-cells and human CD4 T-cells as examples of this:

[REDACTED]

Kalina T, Fišer K, Pérez-Andrés M, Kužílková D, Cuenca M, Bartol SJW, Blanco E, Engel P and van Zelm MC (2019) CD Maps—Dynamic Profiling of CD1–CD100 Surface Expression on Human Leukocyte and Lymphocyte Subsets. *Front. Immunol.* 10:2434. doi: 10.3389/fimmu.2019.02434

[REDACTED]

Song, C., Zhang, L., Wu, X. *et al.* CD4⁺CD38⁺ central memory T cells contribute to HIV persistence in HIV-infected individuals on long-term ART. *J Transl Med* **18**, 95 (2020). doi: 10.1186/s12967-020-02245-8

Minor:

9. Please add the defining clinical characteristics for pneumonia and hypoxia. For hypoxia it seems that an FiO₂ of 40% is used, which is not necessarily an indication for admittance of the ICU at least not in other countries. Also add other characteristics of the patients to this description: particularly the presence of thrombo-embolic events.

Answer: We have now clarified this method section on lines 330-335, page 16: “Pneumonia was diagnosed radiologically by interpretation of CXR or CT thorax images. Hypoxia is defined as requirement for supplemental oxygen, which was started if peripheral O₂ saturations (SpO₂) were <94%. Admission to ICU was reserved for those patients requiring [FiO₂] ≥40% or with haemodynamic instability, and included invasive mechanical ventilation when necessary. Incidence of thrombo-embolic and cardiac events are indicated in Supplemental Table 1.”

10. The presence of neutrophils in samples obtained by bronchoscopy is important (ref. 26). Unfortunately, this is based on a not (yet) peer-reviewed study and this should be indicated as such.

Answer: This is now indicated on lines 244 to 247, page 12 to read: “Supporting this hypothesis, a recent study, not yet peer-reviewed, investigating several myeloid populations between circulating PBMCs and the lung lavage of COVID-19 patients showed that granulocytes represent up to 80% of total CD45+ lung infiltrates²⁹.”

11. Line 329: it is of great concern that two different blood collection tubes were used with different anti-coagulants: EDTA and NaHep. It seems that cytokines were measured in EDTA plasma and cells were analyzed in NaHep. This should be clear and conclusions on the correlations of cytokines and neutrophils in the peripheral blood should be more carefully addressed.

Answer: We would like to thank the reviewer for this comment as we realize that the methods could have been clearer. For cells, only the recovered samples were collected from CPT tubes (Sodium Citrate version, we have now corrected the catalog number), all other cells were collected in EDTA vacuettes. For cytokine analysis, all plasma samples were

collected in CPT tubes. Therefore, even though the cell counts were from one tube type and the cytokines came from another, we believe that correlations analyses are still valid.

In addition, since the reviewer raised an interesting point about the tube and anti-coagulant type effect on cells, we investigated this issue in 5 healthy donors by comparing the cell counts and phenotypic markers between blood collected in EDTA vacuette and CPT tubes. Our results showed that cell counts were not affected but that phenotypic markers (apart for CD38+ on T-cells) were severely decreased in CPT tubes (Supplemental Figure 1A). For this reason, we have removed the recovered group (CPT tubes) from the phenotypic analyses in all figures apart for CD38 gMFI on T-cells (see below appended figure).

Supplementary Figure 1: (a) blood collection in CPT tubes affects phenotypic markers but not cell counts (n=5 healthy donors). Paired analysis was performed either with Wilcoxon non-parametric test for cell counts or ratio of paired t-test for gMFI.

The manuscript text has now been modified on lines 113 to 116 to read: “First, we assessed using healthy donor samples, if the different blood collection method for recovered samples affected cell counts or activation markers. We observed that, while the cell count was not affected, expression of activation markers was affected on most cells but not CD38+ on T-cells (Supplemental Figure 1A).” and at lines 203 to 206 page 10 to read: “To validate that the identified populations would be good markers of disease severity, a correlation analysis between analyte levels in available paired plasma samples (from CPT sodium citrate tubes) was performed with the cell counts obtained in FACS (from EDTA vacuette tubes) (Figure 4A, Supplementary Figure 3).”

Additionally, the methods section corresponding to the flow cytometry (cells) has been amended accordingly and now reads on lines 358-362, page 17: “Blood was collected in VACUETTE EDTA tubes (Greiner Bio, #455036) for healthy donors and acute patients, or Cell Preparation Tubes (CPT) (BD, #362761) for recovered patients. 100 µL of whole blood was extracted for each FACS staining panel (Supplementary Table 3).”

The methods section corresponding to the Luminex analysis now reads on lines 365-366, page 17: "When available, the plasma fraction was harvested after 20 minutes centrifugation at 1700 x g of blood collected in BD Vacutainer CPT tubes (BD, #362761)."

12. Line 336: please provide the data that solvent/detergent treatment does not affect the measured cytokines or show the difference in the online supplement.

Answer: The corresponding methods section on lines 370-372, page 17 now reads: "Cytokines detection in Triton-X treatment was compared with untreated samples for healthy donor and was found to be highly correlative for detected cytokines except for sCD40 (Supplemental Figure 7)." We have added the requested data in Supplemental Figure 7, for easier reviewing, Supplemental figure 7 is copied in below:

A

B

analyte_processed	pvalue	r	r ²
MCP-1	5.9475E-31	0.9952926	0.99060737
PDGF-AA	1.0262E-27	0.99211834	0.98429881
IL-5	2.6522E-12	0.98948711	0.97908473
RANTES	1.6258E-16	0.96794197	0.93691166
GRO	5.7217E-16	0.95647902	0.91485211
MDC	6.139E-13	0.9147584	0.83678294
TNFalpha	7.2058E-13	0.91377281	0.83498075
FGF-2	2.642E-08	0.90060808	0.81109492
IP-10	2.2645E-08	0.87479092	0.76525915
IFNgamma	0.00010308	0.80326746	0.64523862
GM-CSF	0.00145789	0.78560667	0.61717785
IL-17A	0.00260713	0.71736501	0.51461255
EGF	0.01210713	0.56285886	0.3168101
PDGF-AB/BB	0.09238846	0.4668489	0.2179479
MIP-1beta	0.04705156	0.44898796	0.20159019
sCD40L	0.63529644	0.11298794	0.01276627

Supplementary Figure 7: Correlation of analytes detected by Luminex with or without Triton-X treatment in healthy donors. (A) Pearson correlation plots. (B) Pearson correlation p, r and r^2 values for the analytes with readings above detection limit.

13. Line 348: please provide the calculations on which the correction factor is obtained. These can be added to the online supplement.

Answer: We have now added this in the methods on lines 380 to 390 page 18 and reads: "Luminex data was generated from four different runs with each run having a number of samples which are common to the first run. For each subsequent run beyond the first, the

mean of the common samples on each of the plates for each analyte was compared to the mean of the same samples in the first run to obtain a correction factor expressed in the following formula:

$$\text{correction_factor} = \frac{\text{mean}(\text{common_sample_concentration_in_run1})}{\text{mean}(\text{common_sample_concentration_in_subsequent_run})}$$
 - This correction factor was computed for each plate and analyte combination in the subsequent runs and added to the observed concentration to get the final normalised concentration. In the event that none of the common samples had concentration within the standard curve, no correction was done.". The correction factors have now been added to the source data file.

14. FACS analysis in PBS can be problematic for neutrophils and monocytes as they avidly bind to bare plastic surfaces. Most people use PBS with added proteins (e.g. albumin) to minimize this behavior. Please indicate that PBS alone did not lead to lower cell counts in the flowcytometer.

Answer: While the reviewer's comment is true for live cells, it is also possible that fixed cells will adhere to bare plastic. To minimize this possibility, the samples were vortexed before and every 3 min during acquisition. This is now clearly indicated in the methods on lines 403 to 405, page 19 to read: "Samples were then acquired without delay, with vortexing before and every 3 min during acquisition to minimize fixed cell adherence to the tubes".

15. Please indicate that the patients were not having co-infections with bacteria.

Answer: We have now added the bacteria co-infection information in Supplementary Table 1. Please note that, as mentioned in the table, all these co-infections were contracted and diagnosed post-flow cytometry acquisition. All the acute patients had no co-infections at the time of blood draw for flow cytometry staining.

Reviewers' Comments:

Reviewer #3:

Remarks to the Author:

REVIEWER COMMENTS

Reviewer #3 (Remarks to the Author):

Carissimo et al present a descriptive study on leukocyte numbers in the blood of COVID-19 patients. They tested the hypothesis that hyper-inflammation plays an important role in COVID19 and that there is a correlation between disease severity and extend of such inflammation. They compared different disease severities with each other and compared the COVID data with those of healthy controls. They conclude on the basis of many ROC curves that the ratio immature neutrophils/VD2 T-cells is (the best) prognostic marker for deterioration of disease. The article would benefit when the following issues are carefully addressed.

Major (random order):

1. The pathogenesis and main cause of death in COVID-19 are associated with aberrant coagulation and (lung) tissue edema. The majority of critical cases are characterized by thrombo-embolic complications in the lung vasculature. Here ACE2 plays an important role and bradykinin metabolism seems critically involved in the disease mechanism. This of course does not rule out that inflammation takes part in the disease process but it should be carefully placed in the total pathogenesis of COVID-19 particularly in an article in a general science journal such as Nature Communications.

Answer: We are grateful for the suggestions to place the manuscript in a broader context. We have now provided more context on coagulopathy in the introduction on lines 65-72, page 4 to read: "The majority of critical cases of COVID-19 are associated with coagulopathy with a high prevalence of thromboembolic events in patients under mechanical ventilation which lead to inclusion of anticoagulation therapies in the standard of care of severe COVID-19 cases 6-8. While the strong inflammatory response to COVID-19 has been proposed to be associated to COVID-19 associated coagulopathy 6, it remains unclear how SARS-CoV-2 infection affects the activation of immune cells and their contribution towards the different severity of disease outcomes."

However, we feel that discussing the virus entry receptor ACE-2 or the potential role of bradykinin metabolism downstream of viral reduction of ACE-2 would be more appropriate in the context of a review article and out of the scope of this study.

Response: Disease severity of COVID-19 seems primarily caused by tissue edema and coagulopathy. The data supporting hyperinflammation in the onset of severe disease is limited. Even the ARDS associated with critical cases is different when compared with classical (inflammation driven) ARDS. The authors now want to convince the readers that there is an immune signal (ratio banded/VD2 Tcells) associated with an apparently non-immune pathogenic event. This should be carefully addressed rather than stating this should be in the context of a review article.

2. Most of the data in the article are not COVID-19 specific, but generally seen in critical illness. In fact, the data fit with the well-known left-shift in neutrophils in all kind of severe diseases. The data really need bench marking with other patients with other systemic inflammatory conditions. So more left shift (more progenitors, banded cells) in more severe disease is something to be expected. If this is not possible indicate the reason why and at least discuss this possibility in the discussion section.

Answer: We wish to stress that the identification of high immature neutrophil counts in our study specifically showed that in COVID-19 patients there is an accumulation of immature forms of neutrophils which is consistent with other studies. Furthermore, we show that the neutrophil-to-lymphocyte ratio (NLR) proposed by various groups to predict disease outcome in COVID-19 could be improved by looking at immature counts and VD2 T-cells (or CD8) instead to increase the sensitivity and accuracy of this severity indicator. Unfortunately, we do not have access to other cohorts to benchmark this possibility during the current pandemic. Nonetheless, we note the reviewer's point and have discussed this limitation in the revised manuscript on lines 259 to 267, page 12-13 to read: "Interestingly, the diagnostic value of a neutrophil "left shift" (banded versus segmented neutrophils) had previously been explored in order to predict infectious diseases in addition to inflammatory diseases 34 and is therefore not limited to COVID-19 severity. Similarly, the presence of immature low density neutrophils have been reported in the literature for various infectious and inflammatory diseases 35-37 as well as induced by LPS injection in healthy subjects 38, highlighting the necessity of future studies to compare the role and function of these COVID-19 immature neutrophils with the circulating immature neutrophils present in other diseases."

Response: so the authors agree that a left-shift is not specific for COVID-19 but is a measure for severe disease (both inflammatory and tissue damage driven). Now a critical question needs to be answered: is the increased ratio the consequence of severe non-immune disease or is it part of the cause of disease. If it is the consequence than it seems one of the many markers that are different in patients with severe disease.

3. The most important figure seems to be figure 5B. However, this is not easy to understand and needs more information.

- For a marker to be predictive, ROC analysis is the first step but is not sufficient. The next step is to show in a prospective study in new population of patients that the marker is predictive. It is clear that this is impossible now as the number of patients is dropping. So please refrain from using the term predictive in the title of the article.

Answer: We have removed the term prognostic from the title. In addition, we have included the following statement in the discussion: "This prognostic possibility needs to be validated in a prospective cohort." on lines 314 to 315, page 15.

Response: okay

- The authors use a 'median 3 days pio' in their figure and mention an median admittance to the hospital 7 days pio. This really needs explanation how this is done. It is highly likely that there is a selection bias towards patients with an early pio sample available. What is the spread in pio of the two groups? The different times between disease onset and analysis is not appropriate. The need for a prospective cohort is now even more imminent.

Answer: We wish to stress that the days pio refers to the days post illness onset (onset self-reported by the patients) at which the FACS sample was acquired. Patient symptoms information and days pio were made available after flow cytometry analysis. The group that is referred to as median 7 days pio is the full acute patient dataset. Median pio was calculated by doing the median of all datapoints for the acute patients (median and inter quartile range now included in the supplementary table 1). The group at median 3 days pio (Figure 5B) is part of the previous group that fits the following criteria (days pio at time of FACS acquisition of 7 days pio or lower, median was calculated similarly). This is now rephrased for clarity in the manuscript on lines 229 to 232, page 11 to read: "To assess if this analysis could have predictive prognostic value in hospitalisation settings to improve patient

management, we repeated the same analysis with only samples that were acquired at or before 7 days pio amongst the 54 acute patients (24 patients, median pio = 3 days, range 1 to 7 days pio)". We have also modified the Figure 5 B legends to read: "Similar analysis was performed on a subset of early samples from the 54 acute patients (24 patients, 1 to 7 days pio with a median of 3 days pio)."

In addition, we have now added the following statement in the discussion on lines 314-315, page 15: "This prognostic possibility needs to be validated in a prospective cohort."

Response: it is still rather kryptic, but it is good that the authors agree that it all needs validation in a prospective cohort.

4. There seem important inconsistencies in the article that need attention and explanation:

- healthy donors are not age matched and this should be carefully addressed.

Answer: We wish to clarify that there are no inconsistencies, but the fact that the healthy donors are not aged matched is a limitation. We wish to stress that collection of controls was only possible through internal recruitment during the lockdown period in Singapore which prevented us from collecting aged/sex-matched controls. We now discuss this limitation on lines 309 to 310, page 14 to read: "Therefore, it is important to interpret our results between healthy donors and acute samples with the age and sex bias in mind (Supplementary Tables 1 and 2)."

Response: this is not very satisfactory. It would be good for the readers of Nat. Comm. to know whether in the elderly the ratio is similar or different compared to healthy young controls. This is important as age is a very important risk factor in COVID-19. How can readers interpret the data with "age and sex bias in mind"?

- Figure 1: it should really be explained why the recovered patients (time of recovery is 30 days) do not normalize their neutrophil numbers? Are these paired samples? If not, please provide detailed demographics. Otherwise, these data are difficult to interpret.

Answer: We wish to reiterate that, as mentioned in the manuscript, these are not paired samples, detailed demographics are provided in Supplementary Table 1. We are unsure as to what the reviewer is referring to, the recovered patient samples showed similar neutrophil counts as healthy donors since no statistical significance was found between healthy donors and recovered patients in total neutrophils counts (Figure 1F) or in immature and mature counts (Fig1H). Of note, for recovered patients the median pio is 30 days, but the range is 11 to 50 days pio with and IQR of 22.5-35.8 days pio (information now added in Supplementary Table 1).

Response: Okay, I was clearly wrong.

- Figure 1D/E: the use of a relative Z-score in pseudo-colors for single neutrophil activation markers that is difficult to interpret in terms quantification of activation.

Answer: We have now replaced this representation with an interleaved box plot showing all data points for acute versus healthy donors. For ease of reviewing, the new Figures 1D and E are appended here:

Figure 1: SARS-CoV-2 infection induces a decrease in immune cells in peripheral blood. (e) Neutrophil activation markers mean geometric MFI (gMFI) represented as interleaved min to max box plot with all data points shown. (f) Absolute neutrophil counts. (g) Representative plot of mature and immature

neutrophil gating strategy in healthy control or acute COVID-19 patient.

Response: okay

- Figure 1G: this 'representative' figure suggests that almost all neutrophils are CD10⁻ in acute COVID-19 patients. Figure 1H clearly shows that this is not the case. Please avoid such confusion.

Answer: We wish to clarify that the aim of this panel was to illustrate an ideal gating of mature and immature neutrophils using the CD16 and CD10 markers (after CD3⁻ CCR3⁻ CD49d⁻ CD11b⁺ CD66b⁺ gating). We now provide a different flow cytometry plot for Fig1G which has a mix of mature and immature populations, see the new Figure 1G panel below:

Figure 1: SARS-CoV-2 infection induces a decrease in immune cells in peripheral blood. (g)
Representative plot of mature and immature neutrophil gating strategy in healthy control or acute COVID-19 patient.

Response: okay

- It is not clear whether mean or median is depicted in Figure F and H. It seems like the mean is depicted whereas the data is not normally distributed

Answer: All individual plots are represented as scatter dot plots with mean \pm SD. This is now clearly indicated in each figure legend that contains scatter dot plots.

Response: okay

- The data of 1F and 1H imply a 'normal' left shift often seen in severe disease.

Answer: Yes, the data indeed implies a shift from mature neutrophils to immature neutrophils.

Response: okay

- Figure 2C see comment 1D/E.

Answer: In this particular instance, we believe that the heatmap representation allows for a synthetic representation of the changes in frequency instead of the 16 individual plots representation shown in Supplementary Figure 2. We do not believe that moving the 16 plots to Figure 2 will improve the representation.

Response: okay

5. Also figure 3E implies a 'normal' left shift. So the data of figures 1D/E and 3E basically need a comparison with other severe (inflammatory?) PCR-negative disease samples (see comment 1).

Answer: Indeed, this possibility and interesting comparisons are extensively discussed in the manuscript (see previous responses to comment 2). Unfortunately, we do not have access to such cohorts in the current pandemic context.

Response: okay

6. Figure 1D and 4B imply that in COVID-19 up to 90% of the neutrophils are immature in very high numbers. The data in fact show a low expression of CD10. How do the authors know the cells are immature other than low CD10 expression? Hidalgo et al (Trends in Immunology 2019) published a strategy on identifying progenitors on the basis of expression of CD11b and CD16 that was supported by cytopins. Therefore, supportive data of cytopins will show whether the cells have immature characteristics: banded or round nuclei, dark cytoplasm etc. or that something is else is happening

Answer: We understand the reviewer's interrogation. Indeed, in the review article of Hidalgo et al., 2019, the gating strategy for immature subsets of neutrophils relies on CD16 and CD11b expression to identify neutrophil progenitors and precursors in human bone marrow samples. This strategy specifically requires CD62L and CD16 to discriminate between immature banded neutrophils (CD62L+CD16+) and mature poly-segmented neutrophils (CD16++CD62L+) in healthy donor bone marrow or cord blood. However, it is known that CD62L shedding occurs during neutrophil activation which makes the identification of immature neutrophils difficult during inflammation. Moreover, CD16 also upregulates upon activation (See Fig1E of revised manuscript) which would make identifying immature neutrophils even more difficult. Immature neutrophils can be more easily discriminated using other markers such as CD10, which is used in our gating strategy and is in agreement with the review article mentioned by the reviewer.

The portion of this review article by Hidalgo et al., 2019 that the reviewer is referring to, reads: "Flow sorting of the different populations and subsequent analysis of the resulting cytopin preparations demonstrates that it is possible to identify and isolate the different maturing forms of neutrophils in the bone marrow and peripheral blood. Additional markers, such as CD10, CD13, CD64, and CD87, can be used to facilitate the discrimination between mature and immature neutrophils [45-47]."

References 45 to 47 are as follow:

45. Marini, O. et al. (2017) Mature CD10+ and immature CD10- neutrophils present in G-CSF-treated donors display opposite effects on T cells. *Blood* 129, 1343-1356
46. Ng, L.G. et al. (2019) Heterogeneity of neutrophils. *Nat. Rev. Immunol.* 255-265
47. Elghetany, M.T. et al. (2004) Flow cytometric study of neutrophilic granulopoiesis in normal bone marrow using an expanded panel of antibodies: correlation with morphologic assessments. *J. Clin. Lab. Anal.* 18, 36-41"

Specifically, in the study by Marini and colleagues, this morphological confirmation of CD10- and CD10+ neutrophils have been shown with cytopin analyses. The study specifically showed that CD16+CD10- neutrophils contained banded immature phenotypes while CD16+CD10+ neutrophils solely contained mature poly-segmented nuclei phenotypes. Please note that our panel uses a CD16 and CD10 gate after a gate on CD11b and CD66b as well as some exclusion markers. Nevertheless, we had hoped to be able to perform morphology analysis, but the Singapore Ministry of Health Biosafety regulatory board did not allow any workflow to perform blood smears or cytopin analyses on SARS-CoV-2 acute patient samples.

Response: the study by Marini et al used blood of individuals treated with G-CSF for stem cell mobilization. It is very difficult to extrapolate this data such that CD10low/CD16+ are always progenitor/banded cells without morphological confirmation. Surely, this analysis might have been possible on fixed (safe) cells.

7. In Figure 2D the CD38 expression of total CD4 and CD8 is not very useful as it is shown in Figure 2C that the composition of the CD4 and CD8 T cell pool is very different in patients and in Fig 2E that CD38 is very different between the subsets even in healthy controls.

Answer: The reviewer is correct, and we have now removed the illustrative panel 2D from Figure 2

since it had redundant information with the other panels.

Response: okay

8. In Figure 2E it seems odd that CM, EM and TEMRA have lower CD38 gMFI than naïve for both CD4 and CD8 in healthy controls. Also, can the CD38 be reliably measured in CD4 TEMRA while they are almost absent (Sup Fig 2)?

Answer: We agree with this comment, the frequencies and counts of CD4 TEMRA are very low, which renders the measurement of CD38 probably inconsistent. We have now removed this population from the figure and indicated why in the legends. We have removed mention of this population in the manuscript text.

However, for the CD38 expression in other T-cell populations, our results are consistent with several studies indicating that basal CD38 expression is higher in naïve T-cell populations. After differentiation, basal CD38 expression is lower than in naïve T-cells but will re-express CD38 when activated. For this reason, all comparisons were made using healthy donor CD38 gMFI as baseline within the specific CD4, CD8, VD1, VD2 and MAIT T-cell compartments. Please see below the extracted figures from two studies on human T-cells and human CD4 T-cells as examples of this:

Kalina T, Fišer K, Pérez-Andrés M, Kužílková D, Cuenca M, Bartol SJW, Blanco E, Engel P and van Zelm MC (2019) CD Maps—Dynamic Profiling of CD1–CD100 Surface Expression on Human Leukocyte and Lymphocyte Subsets. *Front. Immunol.* 10:2434. doi: 10.3389/fimmu.2019.02434

Song, C., Zhang, L., Wu, X. et al. CD4+CD38+ central memory T cells contribute to HIV persistence in HIV-infected individuals on long-term ART. *J Transl Med* 18, 95 (2020). doi: 10.1186/s12967-020-02245-8

Response: okay

Minor:

9. Please add the defining clinical characteristics for pneumonia and hypoxia. For hypoxia it seems that an FiO₂ of 40% is used, which is not necessarily an indication for admittance of the ICU at least not in other countries. Also add other characteristics of the patients to this description: particularly the presence of thrombo-embolic events.

Answer: We have now clarified this method section on lines 330-335, page 16: "Pneumonia was diagnosed radiologically by interpretation of CXR or CT thorax images. Hypoxia is defined as requirement for supplemental oxygen, which was started if peripheral O₂ saturations (SpO₂) were <94%. Admission to ICU was reserved for those patients requiring [FiO₂] ≥40% or with haemodynamic instability, and included invasive mechanical ventilation when necessary. Incidence of thrombo-embolic and cardiac events are indicated in Supplemental Table 1."

Response: okay

10. The presence of neutrophils in samples obtained by bronchoscopy is important (ref. 26). Unfortunately, this is based on a not (yet) peer-reviewed study and this should be indicated as such.

Answer: This is now indicated on lines 244 to 247, page 12 to read: "Supporting this hypothesis, a recent study, not yet peer-reviewed, investigating several myeloid populations between circulating PBMCs and the lung lavage of COVID-19 patients showed that granulocytes represent up to 80% of total CD45+ lung infiltrates 29."

Response: okay

11. Line 329: it is of great concern that two different blood collection tubes were used with different anti-coagulants: EDTA and NaHep. It seems that cytokines were measured in EDTA plasma and cells were analyzed in NaHep. This should be clear and conclusions on the correlations of cytokines and neutrophils in the peripheral blood should be more carefully addressed.

Answer: We would like to thank the reviewer for this comment as we realize that the methods could have been clearer. For cells, only the recovered samples were collected from CPT tubes (Sodium Citrate version, we have now corrected the catalog number), all other cells were collected in EDTA vacuettes. For cytokine analysis, all plasma samples were collected in CPT tubes. Therefore, even though the cell counts were from one tube type and the cytokines came from another, we believe that correlations analyses are still valid.

In addition, since the reviewer raised an interesting point about the tube and anti-coagulant type effect on cells, we investigated this issue in 5 healthy donors by comparing the cell counts and phenotypic markers between blood collected in EDTA vacuette and CPT tubes. Our results showed that cell counts were not affected but that phenotypic markers (apart for CD38+ on T-cells) were severely decreased in CPT tubes (Supplemental Figure 1A). For this reason, we have removed the recovered group (CPT tubes) from the phenotypic analyses in all figures apart for CD38 gMFI on T-cells (see below appended figure).

Supplementary Figure 1: (a) blood collection in CPT tubes affects phenotypic markers but not cell counts (n=5 healthy donors). Paired analysis was performed either with Wilcoxon non-parametric test for cell counts or ratio of paired t-test for gMFI.

The manuscript text has now been modified on lines 113 to 116 to read: "First, we assessed using healthy donor samples, if the different blood collection method for recovered samples affected cell counts or activation markers. We observed that, while the cell count was not affected, expression of activation markers was affected on most cells but not CD38+ on T-cells (Supplemental Figure 1A)." and at lines 203 to 206 page 10 to read: "To validate that the identified populations would be good markers of disease severity, a correlation analysis between analyte levels in available paired plasma samples (from CPT sodium citrate tubes) was performed with the cell counts obtained in FACS (from EDTA vacuette tubes) (Figure 4A, Supplementary Figure 3)."

Additionally, the methods section corresponding to the flow cytometry (cells) has been amended accordingly and now reads on lines 358-362, page 17: "Blood was collected in VACUETTE EDTA tubes (Greiner Bio, #455036) for healthy donors and acute patients, or Cell Preparation Tubes (CPT) (BD, #362761) for recovered patients. 100 μ L of whole blood was extracted for each FACS staining panel (Supplementary Table 3)."

The methods section corresponding to the Luminex analysis now reads on lines 365-366, page 17:

"When available, the plasma fraction was harvested after 20 minutes centrifugation at 1700 x g of blood collected in BD Vacutainer CPT tubes (BD, #362761)."

Response: okay although it does not make the article much clearer, because now the choice of the anticoagulant affects the conclusion reached in e.g. figure 1E.

12. Line 336: please provide the data that solvent/detergent treatment does not affect the measured cytokines or show the difference in the online supplement.

Answer: The corresponding methods section on lines 370-372, page 17 now reads: "Cytokines detection in Triton-X treatment was compared with untreated samples for healthy donor and was found to be highly correlative for detected cytokines except for sCD40 (Supplemental Figure 7)." We have added the requested data in Supplemental Figure 7, for easier reviewing, Supplemental figure 7 is copied in below:

Supplementary Figure 7: Correlation of analytes detected by Luminex with or without Triton-X treatment in healthy donors. (A) Pearson correlation plots. (B) Pearson correlation p, r and r² values for the analytes with readings above detection limit.

Response: okay

13. Line 348: please provide the calculations on which the correction factor is obtained. These can be added to the online supplement.

Answer: We have now added this in the methods on lines 380 to 390 page 18 and reads: "Luminex data was generated from four different runs with each run having a number of samples which are common to the first run. For each subsequent run beyond the first, the mean of the common samples on each of the plates for each analyte was compared to the mean of the same samples in the first run to obtain a correction factor expressed in the following formula:

$$\text{correction_factor} = \frac{\text{mean}(\text{common_sample_concentration_in_run1})}{\text{mean}(\text{common_sample_concentration_in_subsequent_run})}$$
 This correction factor was computed for each plate and analyte combination in the subsequent runs and added to the observed concentration to get the final normalised concentration. In the event that none of the common samples had concentration within the standard curve, no correction was done."

The correction factors have now been added to the source data file.

Response: okay

14. FACS analysis in PBS can be problematic for neutrophils and monocytes as they avidly bind to bare plastic surfaces. Most people use PBS with added proteins (e.g. albumin) to minimize this behavior. Please indicate that PBS alone did not lead to lower cell counts in the flowcytometer.

Answer: While the reviewer's comment is true for live cells, it is also possible that fixed cells will adhere to bare plastic. To minimize this possibility, the samples were vortexed before and every 3 min during acquisition. This is now clearly indicated in the methods on lines 403 to 405, page 19 to read: "Samples were then acquired without delay, with vortexing before and every 3 min during acquisition

to minimize fixed cell adherence to the tubes”.

Response: okay

15. Please indicate that the patients were not having co-infections with bacteria.

Answer: We have now added the bacteria co-infection information in Supplementary Table 1. Please note that, as mentioned in the table, all these co-infections were contracted and diagnosed post-flow cytometry acquisition. All the acute patients had no co-infections at the time of blood draw for flow cytometry staining.

Response: okay

Reviewer #3 (Remarks to the Author):

Carissimo et al present a descriptive study on leukocyte numbers in the blood of COVID-19 patients. They tested the hypothesis that hyper-inflammation plays an important role in COVID19 and that there is a correlation between disease severity and extend of such inflammation. They compared different disease severities with each other and compared the COVID data with those of healthy controls. They conclude on the basis of many ROC curves that the ratio immature neutrophils/VD2 T-cells is (the best) prognostic marker for deterioration of disease. The article would benefit when the following issues are carefully addressed.

Major (random order):

1. The pathogenesis and main cause of death in COVID-19 are associated with aberrant coagulation and (lung) tissue edema. The majority of critical cases are characterized by thrombo-embolic complications in the lung vasculature. Here ACE2 plays an important role and bradykinin metabolism seems critically involved in the disease mechanism. This of course does not rule out that inflammation takes part in the disease process but it should be carefully placed in the total pathogenesis of COVID-19 particularly in an article in a general science journal such as Nature Communications.

Answer: We are grateful for the suggestions to place the manuscript in a broader context. We have now provided more context on coagulopathy in the introduction on lines 65-72, page 4 to read:

“The majority of critical cases of COVID-19 are associated with coagulopathy with a high prevalence of thromboembolic events in patients under mechanical ventilation which lead to inclusion of anticoagulation therapies in the standard of care of severe COVID-19 cases ⁶⁻⁸. While the strong inflammatory response to COVID-19 has been proposed to be associated to COVID-19 associated coagulopathy ⁶, it remains unclear how SARS-CoV-2 infection affects the activation of immune cells and their contribution towards the different severity of disease outcomes in patients.”

However, we feel that discussing the virus entry receptor ACE-2 or the potential role of bradykinin metabolism downstream of viral reduction of ACE-2 would be more appropriate in the context of a review article and out of the scope of this study.

Response: Disease severity of COVID-19 seems primarily caused by tissue edema and coagulopathy. The data supporting hyperinflammation in the onset of severe disease is limited. Even the ARDS associated with critical cases is different when compared with classical (inflammation driven) ARDS. The authors now want to convince the readers that there is an immune signal (ratio banded/VD2 Tcells) associated with an apparently non-immune pathogenic event. This should be carefully addressed rather than stating this should be in the context of a review article.

Answer 2: We wish to reiterate that in this study, we present the immune phenotyping data from whole blood during SARS-CoV-2 infection in patients with a wide range of symptoms from mild to severe. We showed that several cell types, important for correct immune responses decrease in the circulation, while immature neutrophils drastically increased. These observations correlated closely with the disease severity experienced by the patients.

We strongly believe that these effects are immune related and we postulate, since we could not perform functional testing in patients, that the drastic increase of immature neutrophils reflects a immune-pathogenic event. However, as explained previously, discussing the virus entry receptor ACE-2 or the potential role of bradykinin metabolism downstream of viral action on ACE-2 would be more appropriate in the context of a review article, or in an article on these particular mechanisms rather than in this study on immune cell phenotyping.

2. Most of the data in the article are not COVID-19 specific, but generally seen in critical illness. In fact, the data fit with the well-known left-shift in neutrophils in all kind of severe diseases. The data really need bench marking with other patients with other systemic inflammatory conditions. So more left shift (more progenitors, banded cells) in more severe disease is something to be expected. If this is not possible indicate the reason why and at least discuss this possibility in the discussion section.

Answer: We wish to stress that the identification of high immature neutrophil counts in our study specifically showed that in COVID-19 patients there is an accumulation of immature forms of neutrophils which is consistent with other studies. Furthermore, we show that the neutrophil-to-lymphocyte ratio (NLR) proposed by various groups to predict disease outcome in COVID-19 could be improved by looking at immature counts and VD2 T-cells (or CD8) instead to increase the sensitivity and accuracy of this severity indicator. Unfortunately, we do not have access to other cohorts to benchmark this possibility during the current pandemic. Nonetheless, we note the reviewer's point and have discussed this limitation in the revised manuscript on lines 259 to 267, page 12-13 to read:

“Interestingly, the diagnostic value of a neutrophil “left shift” (banded versus segmented neutrophils) had previously been explored in order to predict infectious diseases in addition to inflammatory diseases⁹ and is therefore not limited to COVID-19 severity. Similarly, the presence of immature low density neutrophils have been reported in the literature for various infectious and inflammatory diseases¹⁰⁻¹² as well as induced by LPS injection in healthy subjects¹³, highlighting the necessity of future studies to compare the role and function of these COVID-19 immature neutrophils with the circulating immature neutrophils present in other diseases.”.

Response: so the authors agree that a left-shift is not specific for COVID-19 but is a measure for severe disease (both inflammatory and tissue damage driven). Now a critical question needs to be answered: is the increased ratio the consequence of severe non-immune disease or is it part of the cause of disease. If it is the consequence than it seems one of the many markers that are different in patients with severe disease.

Answer 2: Yes, we agree that it is a very important question in the current COVID-19 research to understand the cause or consequence in order to better treat patients. Our data, with the experimental limitations in mind, suggest two things: first this ratio, which as agreed with the reviewer might not be COVID-19 specific, seems a much better and earlier marker to predict future severity of symptoms in acute infected COVID-19 patients. Second, it strongly suggests prolonged and sustained neutrophil mobilization and activation in the patients, which is probably pathogenic. This is supported by others in a preprint showing the strong association of low density neutrophils with hypercoagulable state and disease severity in COVID-19 patients (1). In addition, the 5th of august, two studies in Cell which had not been released before as preprints confirm the immature neutrophil and pathogenic

profile of the low density neutrophils using mass cytometry, scRNAseq and whole blood flow cytometry (2-3).

1. Morrissey, S. M., *et al.*, Emergence of Low-density Inflammatory Neutrophils Correlates with Hypercoagulable State and Disease Severity in COVID-19 Patients. *Preprint at <https://www.medrxiv.org/content/medrxiv/early/2020/05/26/2020.05.22.20106724.full.pdf>*
2. Schulte-Schrepping, J., *et al.*, DeutscheCOVID-19 OMICS Initiative (DeCOI), *Severe COVID-19 is marked by a dysregulated myeloid cellcompartment, Cell (2020)*, doi: <https://doi.org/10.1016/j.cell.2020.08.001>.
3. Silvin, A. *et al.*, *Elevated calprotectin and abnormal myeloid cell subsets discriminate severe from mild COVID-19, Cell (2020)*, doi: <https://doi.org/>.

This information on the latested two studies has now been added in the discussion on lines 259 to 261 to read: “More recently, two studies used flow cytometry, single cell sequencing and mass cytometry to confirm the immature and dysfunctional phenotype in the myeloid populations, including these neutrophils ^{34,35}.”

3. The most important figure seems to be figure 5B. However, this is not easy to understand and needs more information.

- For a marker to be predictive, ROC analysis is the first step but is not sufficient. The next step is to show in a prospective study in new population of patients that the marker is predictive. It is clear that this is impossible now as the number of patients is dropping. So please refrain from using the term predictive in the title of the article.

Answer: We have removed the term prognostic from the title. In addition, we have included the following statement in the discussion: “This prognostic possibility needs to be validated in a prospective cohort.” on lines 314 to 315, page 15.

Response: okay

Answer 2: NA

- The authors use a ‘median 3 days pio’ in their figure and mention an median admittance to the hospital 7 days pio. This really needs explanation how this is done. It is highly likely that there is a selection bias towards patients with an early pio sample available. What is the spread in pio of the two groups? The different times between disease onset and analysis is not appropriate. The need for a prospective cohort is now even more imminent.

Answer: We wish to stress that the days pio refers to the days post illness onset (onset self-reported by the patients) at which the FACS sample was acquired. Patient symptoms information and days pio were made available after flow cytometry analysis. The group that is referred to as median 7 days pio is the full acute patient dataset. Median pio was calculated by doing the median of all datapoints for the acute patients (median and inter quartile range now included in the supplementary table 1).

The group at median 3 days pio (Figure 5B) is part of the previous group that fits the following criteria (days pio at time of FACS acquisition of 7 days pio or lower, median was calculated similarly). This is now rephrased for clarity in the manuscript on lines 229 to 232, page 11 to read: “To assess if this analysis could have predictive prognostic value in hospitalisation settings to improve patient management, we repeated the same analysis with

only samples that were acquired at or before 7 days pio amongst the 54 acute patients (24 patients, median pio = 3 days, range 1 to 7 days pio)”.

We have also modified the Figure 5 B legends to read: “Similar analysis was performed on a subset of early samples from the 54 acute patients (24 patients, 1 to 7 days pio with a median of 3 days pio).”

In addition, we have now added the following statement in the discussion on lines 314-315, page 15: “This prognostic possibility needs to be validated in a prospective cohort.”

Response: it is still rather kryptic, but it is good that the authors agree that it all needs validation in a prospective cohort.

Answer 2: NA

- Figure 1: it should really be explained why the recovered patients (time of recovery is 30 days) do not normalize their neutrophil numbers? Are these paired samples? If not, please provide detailed demographics. Otherwise, these data are difficult to interpret.

Answer: We wish to reiterate that, as mentioned in the manuscript, these are not paired samples, detailed demographics are provided in Supplementary Table 1. We are unsure as to what the reviewer is referring to, the recovered patient samples showed similar neutrophil counts as healthy donors since no statistical significance was found between healthy donors and recovered patients in total neutrophils counts (Figure 1F) or in immature and mature counts (Fig1H). Of note, for recovered patients the median pio is 30 days, but the range is 11 to 50 days pio with and IQR of 22.5-35.8 days pio (information now added in Supplementary Table 1).

Response: Okay, I was clearly wrong.

Answer 2: NA

- Figure 1D/E: the use of a relative Z-score in pseudo-colors for single neutrophil activation markers that is difficult to interpret in terms quantification of activation.

Answer: We have now replaced this representation with an interleaved box plot showing all data points for acute versus healthy donors. For ease of reviewing, the new Figures 1D and E are appended here:

Figure 1: SARS-CoV-2 infection induces a decrease in immune cells in peripheral blood. (e) Neutrophil activation markers mean geometric MFI (gMFI) represented as interleaved min to max box plot with all data points shown. (f) Absolute neutrophil counts. (g) Representative plot of mature and immature neutrophil gating strategy in healthy control or acute COVID-19 patient.

Response: okay

Answer 2: NA

- Figure 1G: this 'representative' figure suggests that almost all neutrophils are CD10- in acute COVID-19 patients. Figure 1H clearly shows that this is not the case. Please avoid such confusion.

Answer: We wish to clarify that the aim of this panel was to illustrate an ideal gating of mature and immature neutrophils using the CD16 and CD10 markers (after CD3⁻ CCR3⁻ CD49d⁻ CD11b⁺ CD66b⁺ gating). We now provide a different flow cytometry plot for Fig1G which has a mix of mature and immature populations, see the new Figure 1G panel below:

Figure 1: SARS-CoV-2 infection induces a decrease in immune cells in peripheral blood. (g) Representative plot of mature and immature neutrophil gating strategy in healthy control or acute COVID-19 patient.

Response: okay

Answer 2: NA

- It is not clear whether mean or median is depicted in Figure F and H. It seems like the mean is depicted whereas the data is not normally distributed

Answer: All individual plots are represented as scatter dot plots with mean \pm SD. This is now clearly indicated in each figure legend that contains scatter dot plots.

Response: okay

Answer 2: NA

- The data of 1F and 1H imply a 'normal' left shift often seen in severe disease.

Answer: Yes, the data indeed implies a shift from mature neutrophils to immature neutrophils.

Response: okay

Answer 2: NA

- Figure 2C see comment 1D/E.

Answer: In this particular instance, we believe that the heatmap representation allows for a synthetic representation of the changes in frequency instead of the 16 individual plots representation shown in Supplementary Figure 2. We do not believe that moving the 16 plots to Figure 2 will improve the representation.

Response: okay

Answer 2: NA

5. Also figure 3E implies a 'normal' left shift. So the data of figures 1D/E and 3E basically need a comparison with other severe (inflammatory?) PCR-negative disease samples (see comment 1).

Answer: Indeed, this possibility and interesting comparisons are extensively discussed in the manuscript (see previous responses to comment 2). Unfortunately, we do not have access to such cohorts in the current pandemic context.

Response: okay

Answer 2: NA

6. Figure 1D and 4B imply that in COVID-19 up to 90% of the neutrophils are immature in very high numbers. The data in fact show a low expression of CD10. How do the authors

know the cells are immature other than low CD10 expression? Hidalgo et al (Trends in Immunology 2019) published a strategy on identifying progenitors on the basis of expression of CD11b and CD16 that was supported by cytopins. Therefore, supportive data of cytopins will show whether the cells have immature characteristics: banded or round nuclei, dark cytoplasm etc. or that something else is happening

Answer: We understand the reviewer's interrogation. Indeed, in the review article of Hidalgo et al., 2019, the gating strategy for immature subsets of neutrophils relies on CD16 and CD11b expression to identify neutrophil progenitors and precursors in human bone marrow samples. This strategy specifically requires CD62L and CD16 to discriminate between immature banded neutrophils (CD62L+CD16+) and mature poly-segmented neutrophils (CD16++CD62L-) in healthy donor bone marrow or cord blood. However, it is known that CD62L shedding occurs during neutrophil activation which makes the identification of immature neutrophils difficult during inflammation. Moreover, CD16 also upregulates upon activation (See Fig1E of revised manuscript) which would make identifying immature neutrophils even more difficult. Immature neutrophils can be more easily discriminated using other markers such as CD10, which is used in our gating strategy and is in agreement with the review article mentioned by the reviewer.

The portion of this review article by Hidalgo et al., 2019 that the reviewer is referring to, reads: "Flow sorting of the different populations and subsequent analysis of the resulting cytopin preparations demonstrates that it is possible to identify and isolate the different maturing forms of neutrophils in the bone marrow and peripheral blood. Additional markers, such as CD10, CD13, CD64, and CD87, can be used to facilitate the discrimination between mature and immature neutrophils [45–47]."

References 45 to 47 are as follow:

45. Marini, O. et al. (2017) Mature CD10+ and immature CD10- neutrophils present in G-CSF-treated donors display opposite effects on T cells. *Blood* 129, 1343–1356
46. Ng, L.G. et al. (2019) Heterogeneity of neutrophils. *Nat. Rev. Immunol.* 255–265
47. Elghetany, M.T. et al. (2004) Flow cytometric study of neutrophilic granulopoiesis in normal bone marrow using an expanded panel of antibodies: correlation with morphologic assessments. *J. Clin. Lab. Anal.* 18, 36–41"

Specifically, in the study by Marini and colleagues, this morphological confirmation of CD10- and CD10+ neutrophils have been shown with cytopin analyses. The study specifically showed that CD16+CD10- neutrophils contained banded immature phenotypes while CD16+CD10+ neutrophils solely contained mature poly-segmented nuclei phenotypes. Please note that our panel uses a CD16 and CD10 gate after a gate on CD11b and CD66b as well as some exclusion markers. Nevertheless, we had hoped to be able to perform morphology analysis, but the Singapore Ministry of Health Biosafety regulatory board did not allow any workflow to perform blood smears or cytopin analyses on SARS-CoV-2 acute patient samples.

Response: the study by Marini et al used blood of individuals treated with G-CSF for stem cell mobilization. It is very difficult to extrapolate this data such that CD10low/CD16+ are always progenitor/banded cells without morphological confirmation. Surely, this analysis might have been possible on fixed (safe) cells.

Answer 2: We believe we have clearly explained this aspect in the last previous rebuttal that the Singapore Ministry of Health Biosafety regulatory board did not allow for any workflow that included sharps (i.e. glass slides for blood smear).

7. In Figure 2D the CD38 expression of total CD4 and CD8 is not very useful as it is shown in Figure 2C that the composition of the CD4 and CD8 T cell pool is very different in patients and in Fig 2E that CD38 is very different between the subsets even in healthy controls.

Answer: The reviewer is correct, and we have now removed the illustrative panel 2D from Figure 2 since it had redundant information with the other panels.

Response: okay

Answer 2: NA

8. In Figure 2E it seems odd that CM, EM and TEMRA have lower CD38 gMFI than naïve for both CD4 and CD8 in healthy controls. Also, can the CD38 be reliably measured in CD4 TEMRA while they are almost absent (Sup Fig 2)?

Answer: We agree with this comment, the frequencies and counts of CD4 TEMRA are very low, which renders the measurement of CD38 probably inconsistent. We have now removed this population from the figure and indicated why in the legends. We have removed mention of this population in the manuscript text.

However, for the CD38 expression in other T-cell populations, our results are consistent with several studies indicating that basal CD38 expression is higher in naïve T-cell populations. After differentiation, basal CD38 expression is lower than in naïve T-cells but will re-express CD38 when activated. For this reason, all comparisons were made using healthy donor CD38 gMFI as baseline within the specific CD4, CD8, VD1, VD2 and MAIT T-cell compartments. Please see below the extracted figures from two studies on human T-cells and human CD4 T-cells as examples of this:

[REDACTED]

Kalina T, Fišer K, Pérez-Andrés M, Kužílková D, Cuenca M, Bartol SJW, Blanco E, Engel P and van Zelm MC (2019) CD Maps—Dynamic Profiling of CD1–CD100 Surface Expression on Human Leukocyte and Lymphocyte Subsets. *Front. Immunol.* 10:2434. doi: 10.3389/fimmu.2019.02434

[REDACTED]

Song, C., Zhang, L., Wu, X. *et al.* CD4⁺CD38⁺ central memory T cells contribute to HIV persistence in HIV-infected individuals on long-term ART. *J Transl Med* **18**, 95 (2020). doi: 10.1186/s12967-020-02245-8

Response: okay

Answer 2: NA

Minor:

9. Please add the defining clinical characteristics for pneumonia and hypoxia. For hypoxia it

seems that an FiO₂ of 40% is used, which is not necessarily an indication for admittance of the ICU at least not in other countries. Also add other characteristics of the patients to this description: particularly the presence of thrombo-embolic events.

Answer: We have now clarified this method section on lines 330-335, page 16: “Pneumonia was diagnosed radiologically by interpretation of CXR or CT thorax images. Hypoxia is defined as requirement for supplemental oxygen, which was started if peripheral O₂ saturations (SpO₂) were <94%. Admission to ICU was reserved for those patients requiring [FiO₂] ≥40% or with haemodynamic instability, and included invasive mechanical ventilation when necessary. Incidence of thrombo-embolic and cardiac events are indicated in Supplemental Table 1.”

Response: okay

Answer 2: NA

10. The presence of neutrophils in samples obtained by bronchoscopy is important (ref. 26). Unfortunately, this is based on a not (yet) peer-reviewed study and this should be indicated as such.

Answer: This is now indicated on lines 244 to 247, page 12 to read: “Supporting this hypothesis, a recent study, not yet peer-reviewed, investigating several myeloid populations between circulating PBMCs and the lung lavage of COVID-19 patients showed that granulocytes represent up to 80% of total CD45+ lung infiltrates²⁹.”

Response: okay

Answer 2: Just for clarification, this study has now been published in JCI therefore we have removed “not yet peer-reviewed,” from the sentence at lines 244 to 247:

Sanchez-Cerrillo, I., Landete, P., Aldave, B., Sanchez-Alonso, S., Sanchez-Azofra, A., Marcos-Jimenez, A., Avalos, E., Alcaraz-Serna, A., de Los Santos, I., Mateu-Albero, T., Esparcia, L., Lopez-Sanz, C., Martinez-Fleta, P., Gabrie, L., Del Campo Guerola, L., de la Fuente, H., Calzada, M. J., Gonzalez-Alvaro, I., Alfranca, A., Sanchez-Madrid, F., Munoz-Calleja, C., Soriano, J. B., Ancochea, J. & Martin-Gayo, E. COVID-19 severity associates with pulmonary redistribution of CD1c+ DC and inflammatory transitional and nonclassical monocytes. *J Clin Invest*, 2020.2005.2013.20100925, doi:10.1172/JCI140335 (2020).

12. Line 336: please provide the data that solvent/detergent treatment does not affect the measured cytokines or show the difference in the online supplement.

Answer: The corresponding methods section on lines 370-372, page 17 now reads: “Cytokines detection in Triton-X treatment was compared with untreated samples for healthy donor and was found to be highly correlative for detected cytokines except for sCD40 (Supplemental Figure 7).” We have added the requested data in Supplemental Figure 7, for easier reviewing, Supplemental figure 7 is copied in below:

Supplementary Figure 7: Correlation of analytes detected by Luminex with or without Triton-X treatment in healthy donors. (A) Pearson correlation plots. (B) Pearson correlation p, r and r^2 values for the analytes with readings above detection limit.

Response: okay

Answer 2: NA

13. Line 348: please provide the calculations on which the correction factor is obtained. These can be added to the online supplement.

Answer: We have now added this in the methods on lines 380 to 390 page 18 and reads: "Luminex data was generated from four different runs with each run having a number of samples which are common to the first run. For each subsequent run beyond the first, the mean of the common samples on each of the plates for each analyte was compared to the mean of the same samples in the first run to obtain a correction factor expressed in the following formula:

$$\text{correction_factor} = \frac{\text{mean}(\text{common_sample_concentration_in_run1})}{\text{mean}(\text{common_sample_concentration_in_subsequent_run})} - 1$$
 This correction factor was computed for each plate and analyte combination in the subsequent runs and added to the observed concentration to get the final normalised concentration. In the event that none of the common samples had concentration within the standard curve, no correction was done.". The correction factors have now been added to the source data file.

Response: okay

Answer 2: NA

14. FACS analysis in PBS can be problematic for neutrophils and monocytes as they avidly bind to bare plastic surfaces. Most people use PBS with added proteins (e.g. albumin) to minimize this behavior. Please indicate that PBS alone did not lead to lower cell counts in the flowcytometer.

Answer: While the reviewer's comment is true for live cells, it is also possible that fixed cells will adhere to bare plastic. To minimize this possibility, the samples were vortexed before and every 3 min during acquisition. This is now clearly indicated in the methods on lines 403 to 405, page 19 to read: "Samples were then acquired without delay, with vortexing before and every 3 min during acquisition to minimize fixed cell adherence to the tubes".

Response: okay

Answer 2: NA

15. Please indicate that the patients were not having co-infections with bacteria.

Answer: We have now added the bacteria co-infection information in Supplementary Table 1. Please note that, as mentioned in the table, all these co-infections were contracted and diagnosed post-flow cytometry acquisition. All the acute patients had no co-infections at the time of blood draw for flow cytometry staining.

Response: okay

Answer 2: NA